# Integrative genome-wide analyses identify novel loci associated with kidney stones and provide insights into its genetic architecture

Xingjie Hao [1,3] ✉, Zhonghe Shao[1,3], Ning Zhang[2,3], Minghui Jiang [1,3], Xi Cao[1], Si Li[1], Yunlong Guan[1] & Chaolong Wang [1] ✉

Kidney stone disease (KSD) is a complex disorder with high heritability and prevalence. We performed a large genome-wide association study (GWAS) meta-analysis for KSD to date, including 720,199 individuals with 17,969 cases in European population. We identified 44 susceptibility loci, including 28 novel loci. Cell type-specific analysis pinpointed the proximal tubule as the most relevant cells where susceptibility variants might act through a tissue-specific fashion. By integrating kidney-specific omics data, we prioritized 223 genes which strengthened the importance of ion homeostasis, including calcium and magnesium in stone formation, and suggested potential target drugs for the treatment. The genitourinary and digestive diseases showed stronger genetic correlations with KSD. In this study, we generate an atlas of candidate genes, tissue and cell types involved in the formation of KSD. In addition, we provide potential drug targets for KSD treatment and insights into shared regulation with other diseases.

Kidney stone disease (KSD), also known as nephrolithiasis, is a common ailment, affecting approximately 10–15% of the world's population, with the incidence on the rise[1–4]. Kidney stones are deposits of minerals (primarily calcium) and salts that collect in the kidneys and may become lodged in the urinary tract, thus causing severe pain[5]. The formation of kidney stones has a multifactorial etiology involving both genetic and environmental factors, such as obesity, dehydration and high-sodium diet[5,6]. The lifetime risk of KSD is about 10–15% in developed countries, while it can be as high as 20–25% in the Middle East[7], potentially due to differences in genetic backgrounds and dietary patterns. KSD is a recurrent disease with a recurrence rate of up to 50% within the first 5 years from the initial stone episode, and 75% over 20 years[3,4]. Family history of kidney stones is a key risk factor for recurrent KSD[5]. In twin studies, KSD heritability has been estimated as 0.46–0.57[8,9].

With advances in high-throughput sequencing techniques, monogenic causes of KSD have been identified in up to 30% of children and 10% of adults who have stones, covering about 35 different genes[10–12]. In addition, four genome-wide association studies (GWASs) of KSD based on European and Japanese populations have identified 21 independent and significant susceptibility loci[13–16]. These genetic studies for the monogenic and polygenic factors of KSD have highlighted several important pathways and molecules, such as calcium-sensing receptor signaling pathway, ions and amino acids[1,4]. Recently, more and more tissue and cell type-specific omics data and GWAS summary statistics have become publicly available[17–22], and it has become possible to integrate multiple omics data and traits to explore the genetic architecture of KSD.

In this study, we conduct a large GWAS for KSD to identify more novel mechanistic candidate loci. We further propose identifying the relevant cell types, prioritizing the candidate genes, repurposing drugs and estimating phenome-wide genetic correlations by integrative analysis. Our final goal is to improve the understanding of KSD's genetic architecture.

[1]Department of Epidemiology and Biostatistics, School of Public Health, Tongji Medical College, Huazhong University of Science and Technology, Wuhan, Hubei 430030, China. [2]Department of Breast and Thyroid Surgery, Union Hospital, Tongji Medical College, Huazhong University of Science and Technology, Wuhan, Hubei 430022, China. [3]These authors contributed equally: Xingjie Hao, Zhonghe Shao, Ning Zhang, Minghui Jiang. ✉e-mail: xingjie@hust.edu.cn; chaolong@hust.edu.cn

## Results

### Identifying 28 novel loci for KSD by GWAS meta-analysis

To expand the understanding of KSD's genetic architecture, we conducted a GWAS meta-analysis of KSD in 17,969 cases and 702,230 controls, including European ancestry data from the UK Biobank (UKB)[21] and FinnGen study[23] (Fig. 1). Details regarding GWAS quality control, sample size and phenotype definition are included in the "Methods" section. For the meta-analysis, the genome inflation factor ($\lambda_{gc}$) was 1.017, and the intercept of the LD score regression (LDSC) was 0.956, indicating minimal inflation due to the population structure[24]. The KSD liability heritability was 0.193 (s.e.m. = 0.013), assuming a prevalence of 15%[2,4].

Using the genome-wide significance cutoff ($P < 5 \times 10^{-8}$), we identified 44 independent loci, including 16 previously identified genome-wide significant KSD susceptibility loci (Fig. 2a). We found 56 independent signals at the 44 loci using stepwise conditional analysis, there were one to six independent signals in each locus (Supplementary Data 1). However, five known loci identified in either the Japanese population or meta-analysis of the European and Japanese populations[13,15] (i.e., *GCKR* at 2p23.3, *POU2AF1* at 11q23.1, *WDR72* at 15q21.3, *SCNN1B* at 16p12.2 and *PTGER1* at 19p13.12) did not reach the genome-wide significance. The P values for the lead variants at these five known loci ranged from $2.80 \times 10^{-6}$ to $7.99 \times 10^{-8}$ (Supplementary Data 2). Among these 44 susceptibility loci, the lead variants at 12 loci were associated with calcium, phosphate or 25 hydroxyvitamin D concentration in serum (Supplementary Data 3 and 4). Notably, the risk allele T of rs17216707 for KSD at the known locus *CYP24A1* was significantly associated with higher calcium (beta = 0.059, $P = 8.12 \times 10^{-76}$), phosphate (beta = 0.025, $P = 6.65 \times 10^{-15}$) and 25 hydroxyvitamin D (beta = 0.038, $P = 3.47 \times 10^{-46}$).

We identified 28 novel loci in the GWAS meta-analysis (Fig. 2a, Table 1 and Supplementary Data 5). When unspecified renal colic was included in the cases, 38 independent loci were identified (Supplementary Fig. 1), including 22 novel loci, all of which have been

identified in above analysis. Therefore, our following integrative analyses were based on the analysis which did not include unspecified renal colic as cases. In the replication study which entailed looking up KSD GWAS summary statistics of the Japanese population, the lead variants at 23 novel loci reached the Bonferroni-corrected P value threshold of $1.79 \times 10^{-3}$ (Table 1). Moreover, the lead variants in eight novel loci passed the suggestive genomic significance threshold at $5 \times 10^{-5}$. We also validated the novel loci in the Michigan Genomics Initiative project, and the available lead variants at 27 novel loci showed similar effect sizes with the same sign direction (Supplementary Data 5 and Supplementary Fig. 2).

Among the 3038 GWAS significant variants, the rare variants tended to have larger effect sizes (Fig. 2b). There were 48 significant protein-coding variants, including 26 missense variants located in 20 unique genes (Supplementary Data 6). In addition, these GWAS significant variants showed enrichment in the intronic and untranslated regions (UTR), while they were depleted in the intergenic and ncRNA exonic regions compared with all analyzed variants (Fig. 2c). Stratified heritability analysis with 24 main functional annotations showed that the heritability was most enriched in the conserved regions (Fig. 2d).

### Tissue and cell type specificity for KSD

We applied three approaches integrating different tissue or cell type-specific annotations to identify the relevant tissues and cell types for KSD. Among 54 tissues from the GTEx v.8 project[25], KSD genetic signals only showed enrichment in genes expressed in the kidney medulla ($P = 2.96 \times 10^{-8}$) and cortex ($P = 2.95 \times 10^{-7}$) (Fig. 3a). Next, we conducted stratified LDSC to pinpoint the cell types relevant to KSD. Among 220 cell type-specific chromatin modification marks, the specific marks in six cell types, including three kidney cell types, were relevant to KSD (Fig. 3b, Supplementary Data 7). We further used human kidney snATAC-seq data to identify the relevant cell types for KSD. We found that KSD associated variants showed the greatest enrichment in the proximal tubule specific chromatin accessible

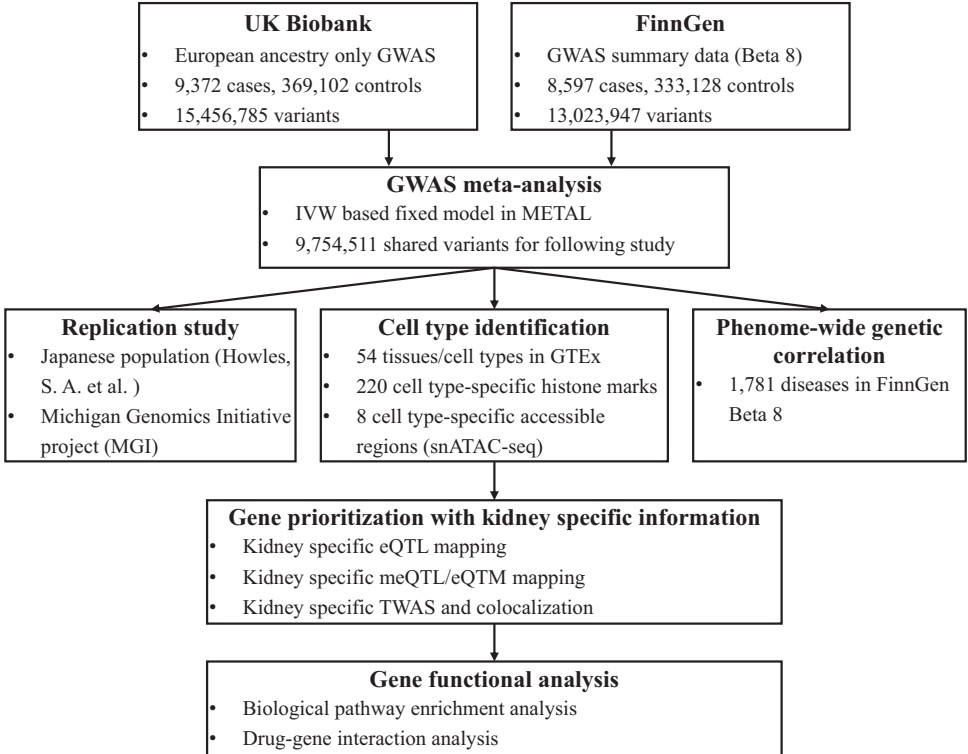

**Fig. 1 | Analysis workflow of the integrative GWAS on kidney stone disease.** Overview of our pipeline for the GWAS meta-analysis and integrative analysis of GWAS hits with multi-omics and multiple traits. See "Methods" and "Results" sections for more details.

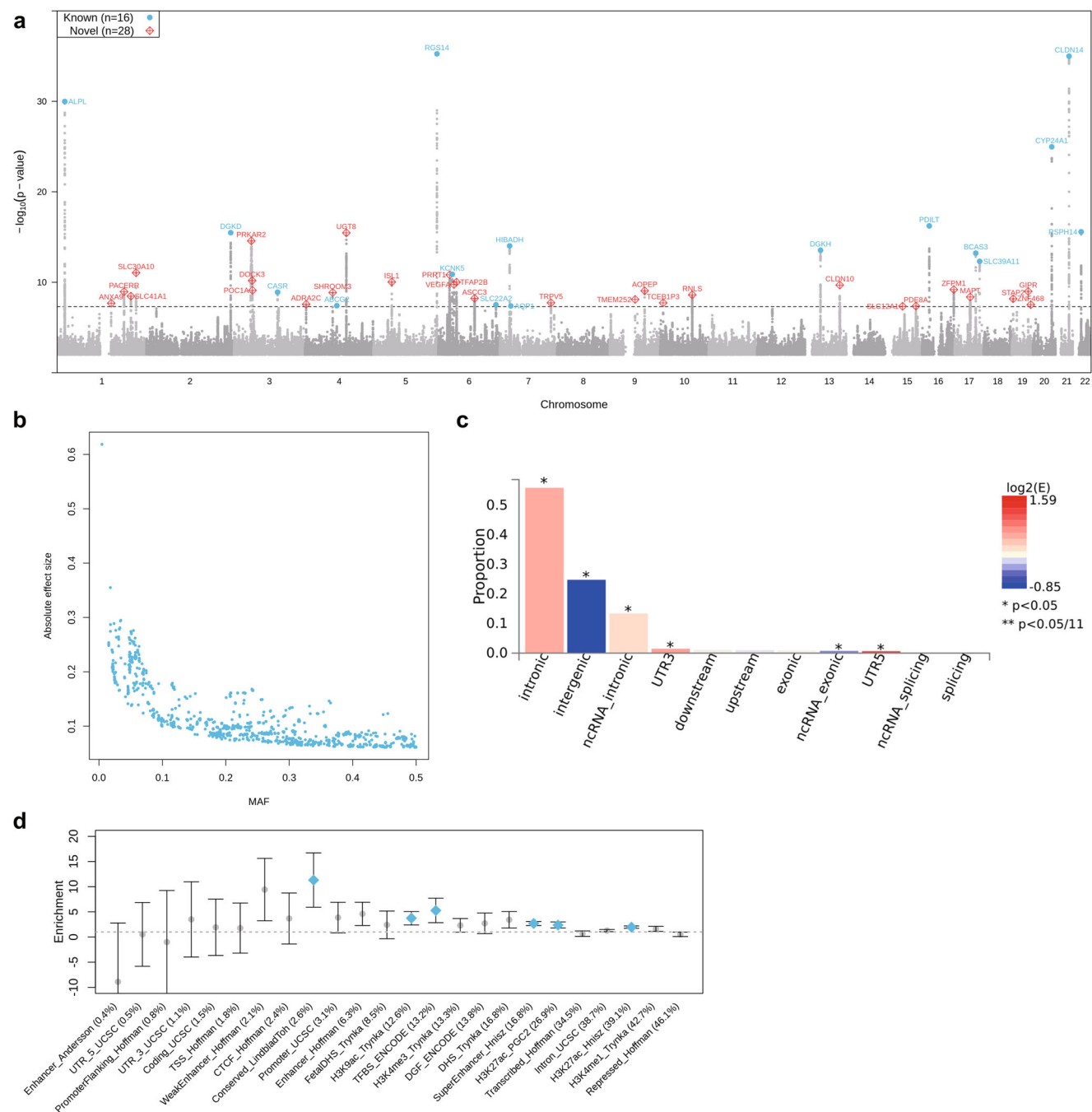

**Fig. 2 | GWAS meta-analysis of kidney stone disease in 720,199 individuals.**
**a** Manhattan plot for the $P$ values of variants. The gray dash line indicates the genome-wide significance level of $5 \times 10^{-8}$. The top variants at the previously identified and novel loci are labeled in blue or red, respectively. **b** Scatter plot for the minor allele frequency (MAF) and absolute effect size of 3038 significant variants. **c** Proportion of GWAS variants in each functional category. Bars are colored by $\log_2$-transformed enrichment value (E; the proportion of GWAS SNPs in a category divided by the proportion of all analyzed variants in the same category).

Asterisks denote significant enrichment or depletion in comparison with all analyzed variants based on Fisher's exact test. **d** Enrichment of SNP heritability in 24 main annotation categories computed by LD score regression. Annotations are ordered by size. Error bars indicate 95% confidence intervals. The blue points denote significant enriched annotations after Bonferroni correction for the 24 hypotheses tested. TSS, transcription start site; CTCF, CCCTC binding factor; DGF, digital genomic footprint; TFBS, transcription factor binding site; DHS, DNase I hypersensitivity site.

regions, while there was enrichment in neither immune cells nor endothelial cells (Fig. 3c).

### Gene prioritization for KSD

To identify the candidate causal genes for KSD, we integrated kidney-specific multi-omics data, including expression quantitative trait locus (eQTL), methylation quantitative trait locus (meQTL) and expression quantitative trait methylation (eQTM). We applied three approaches, namely kidney-specific eQTL mapping, kidney-specific meQTL/eQTM mapping and kidney-specific TWAS and colocalization (see "Methods" section). A total of 223 unique genes were prioritized (Fig. 4a, Supplementary Data 8–11), among which 14 genes were prioritized by three approaches (Supplementary Data 8). *PM2OD1*, *PRELID1*, *RGS14* and *LRRC37A2* were prioritized by three approaches using all different

**Table 1 | Twenty-eight novel loci identified in the genome-wide meta-analysis of kidney stone disease**

| CHR | Position | Lead SNP | EA/NEA | EAF | OR (95% CI) | P | Gene | Replication |
|---|---|---|---|---|---|---|---|---|
| 1 | 150958836 | rs267733 | A/G | 0.845 | 0.92 (0.89–0.95) | $2.07 \times 10^{-8}$ | ANXA9 | $5.17 \times 10^{-3}$ (rs7551686) |
| 1 | 186651932 | rs113831804 | A/T | 0.022 | 1.24 (1.16–1.33) | $1.09 \times 10^{-9}$ | PACERR | $9.64 \times 10^{-4}$ (rs7516899)* |
| 1 | 205724302 | rs823121 | A/G | 0.438 | 1.07 (1.05–1.09) | $3.48 \times 10^{-9}$ | SLC41A1 | $7.56 \times 10^{-6}$ (rs41264867)* |
| 1 | 220082150 | rs884127 | A/G | 0.404 | 0.92 (0.9–0.95) | $8.93 \times 10^{-12}$ | SLC30A10 | $3.00 \times 10^{-4}$ (rs77145552)* |
| 3 | 48861058 | rs200495345 | A/AT | 0.943 | 1.3 (1.22–1.39) | $2.67 \times 10^{-15}$ | PRKAR2A | $2.09 \times 10^{-1}$ (rs3895736) |
| 3 | 51100168 | rs191107165 | A/G | 0.048 | 0.77 (0.71–0.83) | $6.55 \times 10^{-11}$ | DOCK3 | $3.23 \times 10^{-5}$ (rs72964039)* |
| 3 | 52124388 | rs138789058 | T/C | 0.065 | 0.84 (0.79–0.89) | $8.01 \times 10^{-10}$ | POC1A | $3.23 \times 10^{-5}$ (rs72964039)* |
| 4 | 3744294 | rs440318 | A/G | 0.547 | 0.94 (0.92–0.96) | $2.68 \times 10^{-8}$ | ADRA2C | $1.22 \times 10^{-3}$ (rs35867127)* |
| 4 | 77505067 | rs28454965 | A/G | 0.504 | 0.93 (0.91–0.96) | $1.37 \times 10^{-9}$ | SHROOM3 | $2.97 \times 10^{-4}$ (rs7664160)* |
| 4 | 115498457 | rs71606723 | A/T | 0.717 | 0.9 (0.88–0.93) | $3.45 \times 10^{-16}$ | UGT8 | $5.13 \times 10^{-4}$ (rs9884467)* |
| 5 | 51165567 | rs55672774 | T/C | 0.362 | 1.08 (1.05–1.1) | $9.31 \times 10^{-11}$ | ISL1 | $6.00 \times 10^{-5}$ (rs184643180)* |
| 6 | 32107851 | rs3134962 | A/G | 0.155 | 0.9 (0.87–0.93) | $1.51 \times 10^{-11}$ | PRRT1 | $9.37 \times 10^{-8}$ (rs4947328)* |
| 6 | 43804571 | rs729761 | T/G | 0.296 | 1.08 (1.06–1.11) | $1.79 \times 10^{-10}$ | VEGFA | $1.25 \times 10^{-4}$ (rs137924211)* |
| 6 | 50786008 | rs2206271 | A/T | 0.318 | 1.08 (1.05–1.11) | $9.70 \times 10^{-11}$ | TFAP2B | $1.08 \times 10^{-5}$ (rs11961359)* |
| 6 | 101161812 | rs1039031 | A/G | 0.494 | 1.07 (1.04–1.09) | $5.98 \times 10^{-9}$ | ASCC3 | $1.09 \times 10^{-3}$ (rs12210312)* |
| 7 | 142605221 | rs4252512 | T/C | 0.985 | 0.78 (0.71–0.85) | $1.88 \times 10^{-8}$ | TRPV5 | $1.05 \times 10^{-2}$ (rs11520897) |
| 9 | 71172306 | rs12376362 | A/C | 0.823 | 1.09 (1.06–1.12) | $8.00 \times 10^{-9}$ | TMEM252 | $4.22 \times 10^{-3}$ (rs71503670) |
| 9 | 97585477 | rs150891531 | T/C | 0.086 | 0.88 (0.84–0.91) | $8.73 \times 10^{-10}$ | AOPEP | $7.05 \times 10^{-7}$ (rs10993151)* |
| 10 | 9280394 | rs17486892 | T/C | 0.773 | 1.08 (1.05–1.11) | $1.90 \times 10^{-8}$ | TCEB1P3 | $2.08 \times 10^{-4}$ (rs1243417)* |
| 10 | 90142203 | rs11202736 | A/T | 0.680 | 0.93 (0.91–0.95) | $2.47 \times 10^{-9}$ | RNLS | $1.90 \times 10^{-5}$ (rs11203100)* |
| 13 | 96175396 | rs57719175 | A/G | 0.400 | 0.93 (0.91–0.95) | $2.05 \times 10^{-10}$ | CLDN10 | $4.58 \times 10^{-4}$ (rs61152555)* |
| 15 | 48500263 | rs34819316 | A/G | 0.005 | 0.54 (0.43–0.67) | $4.59 \times 10^{-8}$ | SLC12A1 | $2.52 \times 10^{-4}$ (rs117970606)* |
| 15 | 85719783 | rs10852147 | T/G | 0.319 | 1.07 (1.04–1.09) | $4.35 \times 10^{-8}$ | PDE8A | $4.69 \times 10^{-3}$ (rs4887181) |
| 16 | 88549264 | rs55637647 | C/G | 0.650 | 0.93 (0.91–0.95) | $6.50 \times 10^{-10}$ | ZFPM1 | $3.68 \times 10^{-7}$ (rs8054971)* |
| 17 | 44025888 | rs242559 | A/C | 0.808 | 0.92 (0.89–0.94) | $4.22 \times 10^{-9}$ | MAPT | $2.11 \times 10^{-4}$ (rs197923)* |
| 19 | 4335513 | rs58169740 | T/G | 0.683 | 1.07 (1.05–1.1) | $6.74 \times 10^{-9}$ | STAP2 | $3.55 \times 10^{-4}$ (rs58659609)* |
| 19 | 46181392 | rs1800437 | C/G | 0.227 | 0.92 (0.9–0.95) | $1.02 \times 10^{-9}$ | GIPR | $5.72 \times 10^{-5}$ (rs187967034)* |
| 19 | 53357172 | rs7259073 | T/C | 0.916 | 1.13 (1.08–1.18) | $3.02 \times 10^{-8}$ | ZNF468 | $1.09 \times 10^{-3}$ (rs11667654)* |

The lead SNP is the variant with the smallest P value within each locus. The gene is the nearby protein-coding gene to the lead SNP.
*EA* effect allele, *NEA* non-effect allele, *EAF* frequency of the effect allele, *OR* Odds ratio, *CI* confidence interval.
*suggests the variant with the smallest P value at each novel locus in Japanese population passed the significant threshold (0.05/28).

kidney-specific multi-omics data. *PM20D1* and *LRRC37A2* were located at two novel loci: 1q32.1 (Fig. 4b) and 17q21.31 (Fig. 4c), where nine (Fig. 4d) and eighteen (Fig. 4e) genes were prioritized, respectively. In addition, the two novel loci tended to have more open chromatin areas visually in the proximal tubule (Fig. 4b, c).

The prioritized genes also included many known susceptibility genes[13–15], such as *DGKD* prioritized by kidney-specific eQTL mapping, *CASR* and *DGKH* by kidney-specific meQTL/eQTM mapping, *GCKR* and *SCNN1B* by kidney-specific TWAS, *ALPL*, *SLC34A1*, *HIBADH*, *UMOD* and *CLDN14* by kidney-specific eQTL mapping and meQTL/eQTM mapping, *BCR* prioritized by kidney-specific meQTL/eQTM mapping and TWAS (Supplementary Data 8). In addition, we found that 126 prioritized genes were expressed at protein level, as detected by microarray-based immunohistochemistry in the kidney, including proximal tubules, distal tubules, collecting ducts, cells in tubules, and cells in glomeruli and Bowman's capsule (Supplementary Data 12). Moreover, the prioritized genes showed significant enrichment in 11 gene sets, including "phosphate ion homeostasis" (GO:0055062) (Supplementary Data 13).

### Drug-gene interactions for the prioritized genes

Among all of the prioritized genes, there were 61 unique genes associated with 1534 unique drugs and 1909 drug-gene interaction pairs (Supplementary Data 14). *EHMT2* prioritized by kidney-specific meQTL/eQTM mapping was the top gene, related to 864 drugs, followed by 436 drugs for *MAPT*, 151 for *GLP1R*, 62 for *ADRA2C*, and 51 for

*TUBB*. These drugs were classified into distinct types, including 56 inhibitors, 37 agonists, 4 blockers, 37 antagonists, and 5 modulators. Among the 1534 gene-associated drugs identified, a total of 48 drug-gene interaction pairs were found to be related to KSD, including 38 drugs for calcium renal calculi prevention, 19 for treating or reducing the symptoms of calcium oxalate renal calculi, 16 for calcium phosphate renal calculi, 38 for calcium renal calculi prevention, 9 for urate renal calculi and 2 for struvite renal calculi. Among the drug-gene interaction pairs for *EHMT2*, 24 drugs can be composed of medications for KSD treatment. For instance, Cytra-2, Cytra-3, and Urocit-k which contain the identified drugs (e.g., 2′-Hydroxychalcone, indole-3-carbinol and indirubin-3′-monoxime) can make the urine less acidic, thus helping the kidneys eliminate uric acid, and prevent gout and certain types of kidney stones. Moreover, hydrochlorothiazide is widely used to treat calcium-containing KSD because it decreases the amount of calcium excreted by the kidney in the urine, and thus decreases the amount of calcium in urine to form stones.

### Genetic correlations with other diseases

Genetic correlations were estimated to explore the potential genetic overlap between KSD and 1781 diseases in the FinnGen study[23]. We observed significant genetic correlations with 61 diseases. These significant genetic correlations were all positive and ranged from 0.15 to 1.06 (Fig. 5; Supplementary Data 15). The diseases in the categories of genitourinary, infectious and digestive diseases showed stronger genetic correlations with KSD, where the median genetic correlation

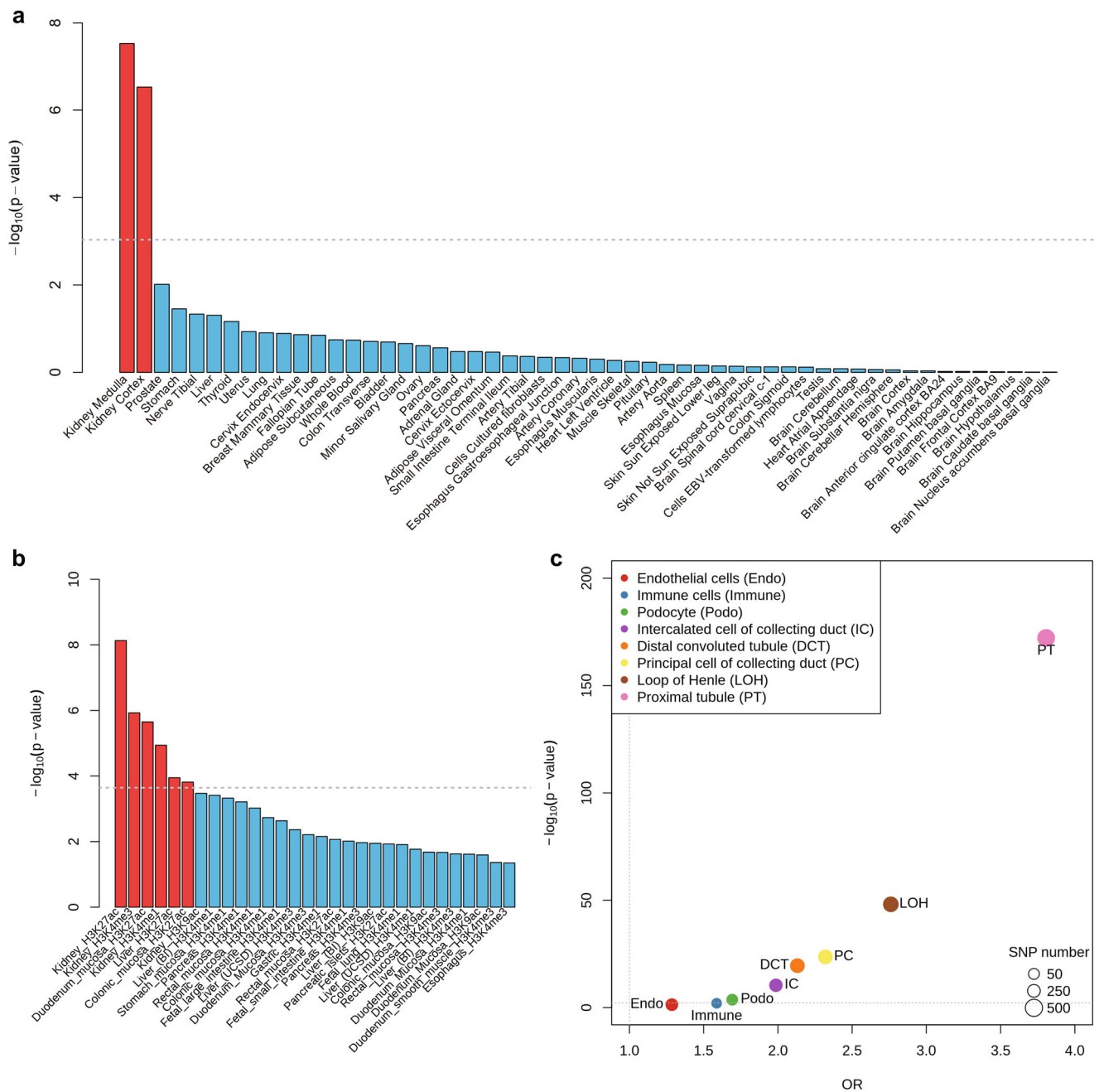

**Fig. 3 | Relevant tissue and cell types for kidney stone disease. a** The relevant tissues for kidney stone disease by MAGMA gene-property analysis. The x-axis is the 54 GTEx tissue-specific gene expression profiles and the y-axis is −log₁₀ *P* values of enrichment test. **b** Enrichment of individual cell types by stratified LD score regression. Thirty cell types with enrichment *P* value smaller than 0.05 are shown here, and the whole 220 cell type enrichment results are in Supplementary Data 7.

**c** Enrichment of kidney stone disease significant variants in kidney cell type-specific accessible regions. The dot size represents the number of variants overlapping with differentially accessible regions in given cell types. Horizontal dashed lines indicate the significance thresholds of **a** 0.00093 (0.05/54), **b** 0.000227 (0.05/220), and **c** 0.00625 (0.05/8) after Bonferroni correction.

coefficients were 0.36, 0.35, and 0.26, respectively, and 0.22 for the diseases in other categories. KSD had the strongest genetic correlation with calculus of lower urinary tract ($r_g = 1.06$, $P = 9.24 \times 10^{-6}$), another stone-related disease in the genitourinary system, followed by cystitis ($r_g = 0.43$, $P = 1.09 \times 10^{-6}$), renal tubulo-interstitial diseases ($r_g = 0.37$, $P = 3.71 \times 10^{-6}$), diarrhea and gastroenteritis of presumed infectious origin ($r_g = 0.37$, $P = 9.88 \times 10^{-6}$), other diseases of urinary system ($r_g = 0.35$, $P = 7.50 \times 10^{-9}$), and other diseases of intestines ($r_g = 0.34$, $P = 2.49 \times 10^{-15}$). We found no significant genetic correlation between KSD and vitamin D deficiency, although they share several susceptibility genes in some potential pleiotropic loci.

## Discussion

We performed a large GWAS meta-analysis for KSD, covering 720,199 individuals. We identified 44 KSD susceptibility loci, including 28 novel loci. Cell type-specific analysis pinpointed the proximal tubule cells in the kidney as the most relevant cell type for KSD. We prioritized the candidate genes for KSD by integrating kidney-specific omics data. In addition, we explored the genetic correlations with other diseases.

In this GWAS meta-analysis, we combined two large studies, UKB[21] and FinnGen[23]. Although GWAS on KSD in UKB has been conducted previously[13], many incident KSD cases have been reported in UKB since then. With more samples, we identified 44 KSD susceptibility loci,

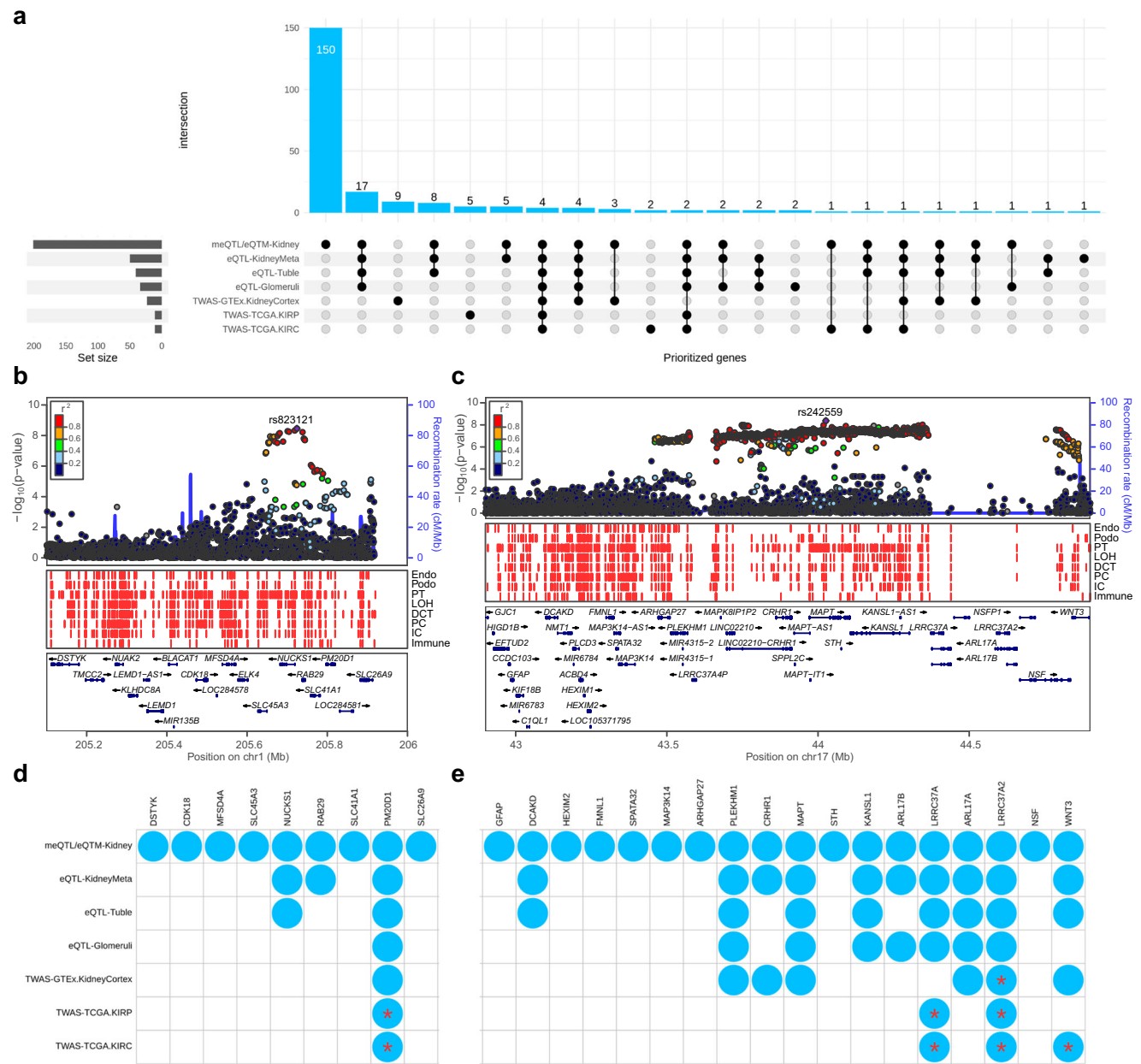

**Fig. 4 | Gene prioritization for kidney stone disease. a** The prioritized genes identified by kidney-specific eQTL and meQTL mapping and TWAS. **b** Regional plots of GWAS signals at 1q32.1. **c** Regional plots of GWAS signals at 17q21.31. The lead variants are indicated by the purple diamond and black labeled, while the neighboring variants are colored based on their LD with the lead variant in European samples from 1000 Genomes Project. Gray color indicates LD information not available in the 1000 Genomes Project. The snATAC-seq chromatin accessible regions of human kidney are labeled in red. **d** The prioritized genes at 1q32.1. **e** The prioritized genes at 17q21.31. The asterisk indicates the GWAS signals are colocalized with the kidney-specific eQTL at the TWAS prioritized genes.

including 16 previously reported loci[13–16]. In addition, we also discovered 28 novel loci and validated 23 novel loci in the Japanese population, indicating that most causal variants were shared across diverse populations. Although we conducted a very large GWAS meta-analysis for KSD, the heritability explained by all SNPs was 0.193 which was only about 34–42% of heritability based on the twin studies[8,9]. These results suggest that future studies involving more samples from diverse ancestry could be promising in detecting more loci for KSD.

Tissue and cell type-specific enrichment analysis supported that the kidney, and especially the proximal tubule, were the key target tissue and cell type for KSD. The proximal tubule has a very low transepithelial resistance and is highly permeable to water and small ions, especially calcium, sodium, and chloride. In fact, the proximal tubule is responsible for the majority of calcium reabsorption in the

kidney[26]. A considerable amount of evidence has suggested a crucial role for the proximal tubule in idiopathic hypercalciuria and the development of KSD[27–29]. The associated signals were the most significantly enriched in the proximal tubule open chromatin regions, suggesting that KSD candidate genes may influence the function of the proximal tubule in a cell type-specific manner. For example, *CYP24A1* had the highest expression in kidney based on GTEx profiles (Supplementary Fig. 3), and there were significant correlations between the risk allele T of rs17216707 for KSD at *CYP24A1* and higher calcium, phosphate and vitamin D concentration. This evidence suggested a potential mechanism whereby the T allele at rs17216707 could increase the risk of KSD through the dysfunction of *CYP24A1*, which could result in the elevated 25 hydroxyvitamin D, further increase the reabsorption of calcium in the kidney. In addition, we also noticed that KSD signals

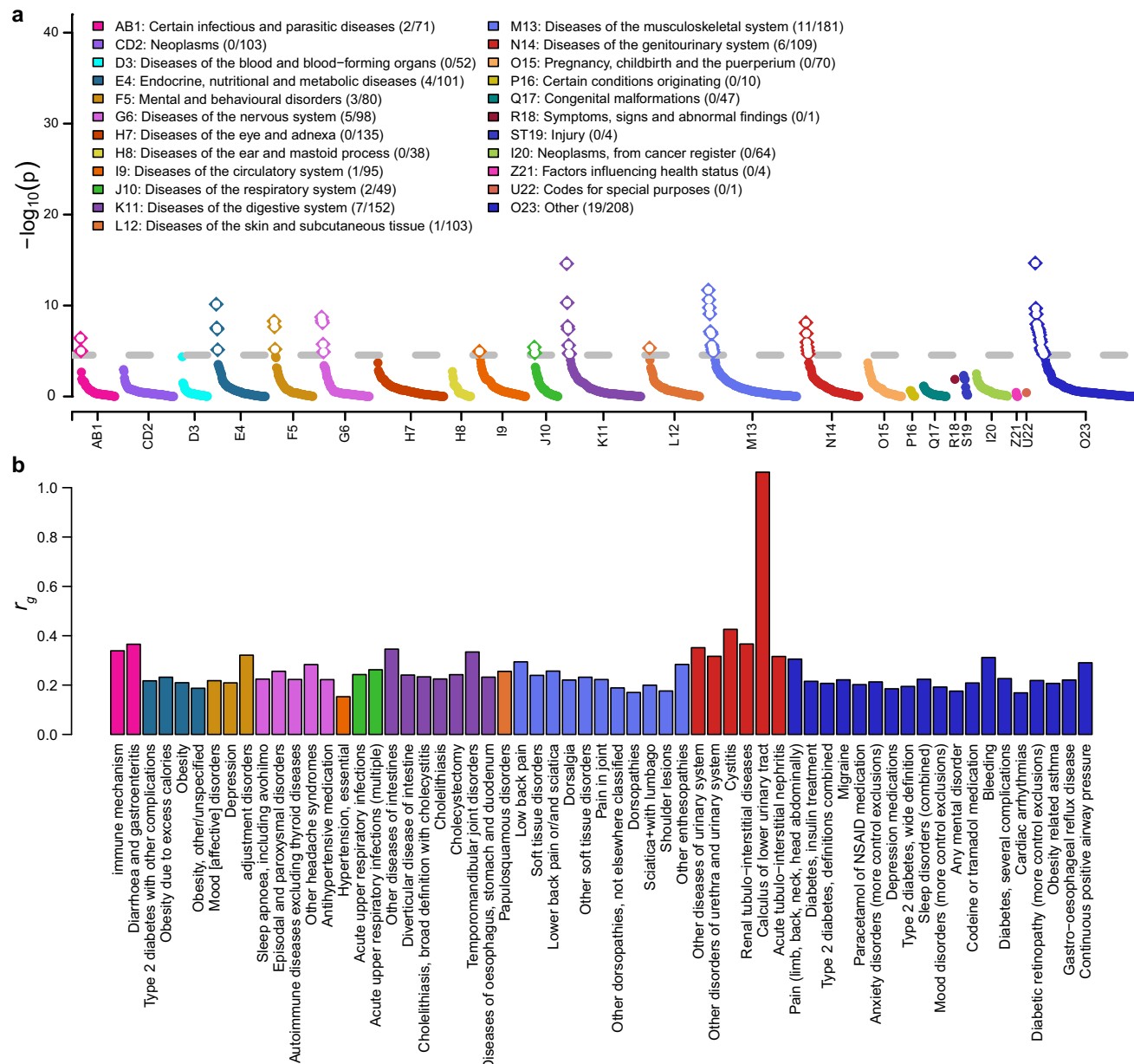

**Fig. 5 | Phenome-wide genetic correlations with kidney stone disease. a** The *P* values of genetic correlations with kidney stone disease. The gray dash line is the significant threshold of $2.81 \times 10^{-5}$ after Bonferroni correction for 1781 diseases. The numbers of significant genetic correlations and diseases in each disease category are in the parentheses. **b** The significant genetic correlations with kidney stone disease. The colors denote different disease categories according to the ICD-10 code.

were enriched in the annotation regions of liver and intestine (Fig. 3b) which were the main organs for the metabolism and absorption of vitamin D, respectively. Vitamin D plays an important role in the absorption of calcium in the intestine[30]. Indeed, both vitamin D and calcium are associated with KSD[13,31,32]. Our tissue and cell type-specific enrichment analysis indicated the kidney stone formation was influenced not only by the metabolism and absorption of vitamin D and calcium in the liver and intestine, but also by the calcium reabsorption in the kidney.

To pinpoint the candidate causal genes for KSD, we integrated the kidney-specific omics data to prioritize genes using different approaches. Among the prioritized genes at the novel locus 1q32.1 (Fig. 4), *PM20D1* is involved in the lipid metabolic process and regulating energy homeostasis[33]. Serum PM20D1 levels are significantly elevated in overweight and obese individuals[34]. Adiposity is well known to

increases risk of KSD[3,4]. Associations of PM20D1 with KSD and obesity may indicate a shared causal pathway. *SLC41A1* was essential for magnesium homeostasis in vivo[35,36] and had a Mendelian association with nephronophthisis-like nephropathy-2 (OMIM phenotype number, #619468) that is the most common genetic cause of chronic kidney disease in early life. Chronic kidney disease could cause metabolic acidosis, resulting in a reduced urine pH, which could further promote the formation of uric acid kidney stones[37]. Additionally, KSD individuals tended to have lower magnesium in their blood than people without kidney stones[38], potentially because the magnesium can inhibit calcium oxalate kidney stones by binding with oxalate, thus making oxalate less likely to bind with calcium to form kidney stones. Moreover, *SLC45A3* and *SLC26A9* were involved in the biological process of ion transport and pH regulation, which was crucial to kidney stone formation[39]. At another novel locus 17q21.31 (Fig. 4), *LRRC37A2* and

*LRRC37A* were predicted to be the integral component of the membrane and involved in protein binding, which may be involved in transmembrane transport. *WNT3* has been shown to play a leading role during bladder development and had been implicated in the pathogenesis of the group of conditions called bladder-exstrophy-epispadias complex[40], which has a profound impact on renal functions. *WNT3* also showed a Mendelian association with tetra-amelia syndrome 1 (#273395), and renal and adrenal agenesis have been observed in tetra-amelia syndrome 1 patients[41]. *PLEKHM1* was involved in the biological process of metal ion binding and associated with osteopetrosis (#618107)[42]. Previous studies have shown elevated parathyroid hormones (PTH) in osteopetrosis patients[43,44]. Additionally, PTH can regulate calcium metabolism by increasing the production of $1,25(OH)_2D$ and calcium reabsorption[45]. Our results suggested a potential shared regulatory mechanism of *PLEKHM1* for osteopetrosis and KSD.

Furthermore, based on position mapping of the novel loci, we identified multiple genes involved in ion transportation (Table 1 and Supplementary Data 5). For example, *SLC30A10* at 1q41 encodes manganese transporter[46], which may regulate manganese concentration to affect the formation of calcium oxalate kidney stones. It also has a Mendelian association with hypermanganesemia with dystonia 1 (#613280)[46]. In addition, previous studies have shown that the manganese concentration in urine is lower in KSD patients[47] and the higher manganese intake is associated with a lower risk of KSD[48]. *CLDN10* at 13q32.1 encodes a member of the claudin family that regulates the transepithelial ion exchange and electrical resistance[49]. *CLDN10* is associated with HELIX syndrome (#617671), as biallelic mutations cause dysfunction in tight junctions in several tissues, and subsequent abnormalities in renal ion transport, ectodermal gland homeostasis, and epidermal integrity[50]. Specifically, previous whole-exome analysis in the UKB suggested that monoallelic deletion in *CLDN10* is related to kidney function and chronic kidney disease[51]. These findings highlighted the role of *CLDN10* in ion transportation which is crucial to KSD. Also, calcium ($Ca^{2+}$) plays a key role in stone formation, and the epithelial $Ca^{2+}$ channel transient receptor potential vanilloid (*TRPV5*) was a long-standing candidate gene for KSD and hypercalciuria[52]. However, no association with $P < 5 \times 10^{-8}$ has previously been identified at this locus, probably due to the small sample size[13–16]. In this study, we identified a novel signal located 46 bp downstream of the *TRPV5* showing an association with KSD (rs4252512, MAF = 0.015, OR = 0.78, $P = 1.88 \times 10^{-8}$), which may explicate the role of *TRPV5* in the pathology of KSD. We identified a rare missense variant, rs34819316 (MAF = 0.005, OR = 0.54, $P = 4.58 \times 10^{-8}$) in *SLC12A1*, which was significantly correlated with KSD (Supplementary Data 6). *SLC12A1* encodes a kidney-specific sodium-potassium-chloride cotransporter, and functions in urine concentration and NaCl reabsorption[53]. Biallelic mutations in *SLC12A1* were associated with hypercalciuria, and may even cause Bartter syndrome (#601678)[54].

Moreover, our results suggested other biological pathways that may be involved in KSD. The most significantly enriched gene sets in the pathway analysis were "phosphate ion homeostasis" (GO:0055062), "limb morphogenesis" (GO:0035108) and "appendage morphogenesis" (GO:0035107). Previous studies have shown that disturbances in phosphate homeostasis can lead to KSD[55]. Among the prioritized genes in "limb morphogenesis" and "appendage morphogenesis" pathways, the *TFAP2B* gene is involved in "renal system development" (GO:0072001) and "kidney development" (GO:0001822). Moreover, *ECE1* is involved in "calcitonin catabolic process" (GO:0010816). In addition, *MED1* is involved in vitamin D signaling pathway, such as "regulation of vitamin D receptor signaling pathway" (GO:0070562), "cellular response to vitamin D" (GO:0071305), and "response to vitamin D" (GO:0033280). These findings suggested that ion homeostasis and kidney-related development play important roles in the formation of KSD.

In the phenome-wide genetic correlation scanning, KSD showed positive genetic correlations with 61 other diseases, and had stronger genetic correlations with genitourinary and digestive diseases, including two infectious digestive diseases (i.e., intestinal infectious diseases, diarrhea and gastroenteritis of presumed infectious origin). KSD is also a type of genitourinary disease and may often co-exists with kidney infection[56]. Thus, it was not surprising to see its stronger genetic correlations with other genitourinary diseases, especially another stone-related disease in the genitourinary system, namely calculus of lower urinary tract. The stronger genetic correlations with genitourinary diseases also suggested shared genetic structures for the comorbidity of other genitourinary diseases with KSD. The genetic correlations with digestive diseases highlighted the gut-kidney axis[57]. The healthy status and homeostasis of the intestine are important to the absorption of vitamin D and calcium, which can influence the formation of kidney stones[13,31,32]. However, the genetic correlation did not represent the causality which can be bidirectional and needs to be inferred through proper methods, such as Mendelian randomization[58].

Several limitations warrant mention. First, although tissue and cell type-specific analysis suggested that the liver and intestine were also relevant to KSD, we only utilized kidney-specific omics data. This is because kidney-specific omics data are comprehensive, in addition, most of which are publicly available for gene prioritization. Second, we applied three approaches to prioritize genes using different omics data, but we did not use several other established approaches, such as colocalization and fine-mapping for the causal variants. This is due to the restriction of full summary statistics for kidney-specific omics data. In addition, the gene prioritization approaches may fail to prioritize genes at some susceptibility loci if the associated variants are not available in the omics data. Third, we did not conduct in vitro experiment for the novel loci or the prioritized genes, even though they were validated in independent samples and supported by multi-omics data from different sources and approaches. Fourth, we provided a list of potential drug targets for KSD treatment without further validation by experiments, which will be required in future studies. For example, a recent double-blind trial suggested that the beneficial effects of hydrochlorothiazide were controversial and might not mitigate the risk of KSD recurrence[59]. Therefore, further studies of hydrochlorothiazide's long-term effect on KSD are needed. Fifth, the genetic correlation analysis was focused on diseases, while we did not explore the quantitative traits, such as estimated glomerular filtration rate, blood calcium and phosphate. In addition, genetic correlations were at a genome-wide scale, but some diseases may only share a few pleiotropic loci or local genetic correlations with KSD.

In conclusion, this large-scale integrative genome-wide analysis of KSD has generated an atlas of candidate genes, tissues and cell types, and pathways involved in the formation of kidney stones. Furthermore, this study also provided a list of the potential drug targets for KSD, and insights into shared genetic regulation with other disease.

## Methods

### UK Biobank genotyping and quality control

The UK Biobank (UKB) cohort is a population-based volunteer longitudinal cohort of ~500,000 individuals recruited at 22 centers across the United Kingdom. We used the imputed genotypes from UKB. Details of the array, sample processing and quality control have been described in detail elsewhere[21]. From the resulting dataset, we extracted a European ancestry subset, including 408,812 samples who self-identified as white British and had very similar genetic ancestry based on the principal component analysis of the genotypes. The variants with minor allele frequency (MAF) < 0.001, Hardy–Weinberg equilibrium test $P$ value $< 1.0 \times 10^{-12}$, missing genotype rate >0.05, or imputation accuracy score <0.3 were excluded by using PLINK[60]. We also randomly remove the individuals in related pairs with kinship coefficient ≥0.0884 (second degree or greater). Finally, 378,474

European ancestry individuals and 1,545,6785 variants on GRCh37 were retained for the following study. The application number of UKB in our study was 88159.

## GWAS on KSD in the UK Biobank
KSD phenotype in UKB was derived from the first occurrence of any code mapped to 3-character ICD-10 (category ID 1712), which was generated by mapping ICD-10 code in the death register records, read code information in the primary care data, ICD-9 and ICD-10 codes in the hospital inpatient records (e.g., diagnosis or operation code), and self-reported medical conditions reported at the baseline or subsequent visit to UKB assessment center. KSD cases were defined as calculus of kidney and ureter (field ID 132036, ICD-10 N20), and the remaining samples were controls. Following quality control, 9372 UKB participants were identified as cases and 369,102 individuals as controls by July, 2022. In addition, we also included the unspecified renal colic (field ID 132042, ICD-10 N23) as the cases, therefore 11,948 participants were identified as cases and 366,526 individuals as controls. We performed GWAS analysis using logistic regression with PLINK[60], adjusting for sex, array batch, and top ten ancestry principal components.

## GWAS summary statistics for KSD from FinnGen study
The FinnGen study includes six regional and three country-wide Finnish biobanks. Participants health outcomes were followed up by linking to the national health registries, which collected information from birth to death[23]. Summary statistics for KSD (data freeze 8, Fall 2021, code N14_CALCUKIDUR) were publicly available (https://r8.finngen.fi/). The 8597 KSD cases in FinnGen were defined as calculus of kidney and ureter (ICD-10 N20), and the remaining samples that further excluded urolithiasis were controls (n = 333,128). The variant positions in FinnGen study were based on GRCh38, and were flipped to GRCh37. The variants with MAF < 0.001 were removed.

## GWAS meta-analysis
We performed the GWAS meta-analysis by using the fixed-effect inverse-variance weighted model implemented by METAL[61] with genomic control correction for each input study. Specifically, 9,754,511 variants shared in UKB and FinnGen were retained for the following study.

We defined variants associated with KSD by genome-wide significance level ($P < 5 \times 10^{-8}$). Independent loci were defined for the associated variants using the following steps. First, we clumped the significant variants using PLINK[60] to choose the lead and independent variants in a locus (commend: –clump-p2 5e-8 --clump-r2 0.1 --clump-kb 10000[62]). The LD information was based on 1,000 randomly selected European samples from UKB. To avoid calling multiple associations for very large signals, lead variants within 1000 kb of each other were merged together. A total of 44 independent loci were identified. In each locus, the variant with the minimum $P$ value was selected to search the nearby gene and represent the locus. Stepwise conditional analysis was conducted to identify additional significant signals using GCTA[63] with the above mentioned LD reference panel. In addition, the variants in the associated loci were annotated using FUMA[64]. Enrichment of associated variants in each annotation was tested by Fisher's exact test (two-sided) by comparing them with the annotations of all variants in the meta-analysis.

To identify novel loci, we compared the 44 independent loci to the previously reported KSD loci in GWAS Catalog by February 2023[22]. Independent loci by our meta-analysis were defined as novel if they did not pass genome-wide cutoff $P < 5 \times 10^{-8}$ in any previous study or were located more than 1000 kb away from any of the previously reported index variants. Finally, we identified 28 novel loci for KSD.

## Replication in the Japanese population
To validate the novel loci identified in our GWAS meta-analysis, we tried to search the signals in KSD GWAS in the Japanese population. We extracted the summary statistics of KSD GWAS in the Japanese population from a previous study which conducted trans-ethnic GWAS meta-analysis of KSD by combining UKB (6536 cases and 388,508 controls) and the Japanese population (5587 cases and 28,870 controls)[13]. GWAS summary statistics for UKB and UKB/Japanese meta-analysis were from https://doi.org/10.5287/bodleian:2NEEgv2QD. We inferred z-score for each variant in the Japanese population using the following formula.

$$ Z_{JP} = \frac{Z_{meta}\sqrt{n_{meta}} - Z_{UKB}\sqrt{n_{UKB}}}{\sqrt{n_{JP}}} $$

In this formula, $Z_{UKB}$ and $Z_{meta}$ indicated z-scores of KSD GWAS in UKB and UKB/Japanese meta-analysis. $n_{UKB}, n_{JP}$ and $n_{meta}$ were the effective sample size for UKB, Japanese and meta-analysis. The $P$ value for each variant in the Japanese population was obtained from $2\Phi(-|Z_{JP}|)$. This formula was derived from the sample size based meta-analysis model, an approach known to be asymptotically equivalent to inverse variance based meta-analysis[61].

The index variant tagging the causal variant could be different in the European and Japanese populations due to the different LD patterns and allele frequency. In addition, many associated variants in our meta-analysis were not available in the summary statistics of KSD GWAS in the Japanese population. We defined the novel locus as validated if any variants in the novel locus in the Japanese population reached a Bonferroni-correlated $P$ value threshold of $1.79 \times 10^{-3}$ which adjusted for 28 novel loci.

## Replication in the Michigan Genomics Initiative
We also conducted the lookup study for the lead variants from the novel loci in the Michigan Genomics Initiative (MGI) which is a biobank of Michigan medicine patients. The MGI PheWeb (https://pheweb.org/MGI/) contained results from GWASs of 1542 EHR-derived phenotypes for approximately 51.8 million imputed variants in 51,583 European ancestry individuals. Specifically, the KSD GWAS in MGI contained 6358 cases and 43,669 controls.

## Tissue and cell type identification
We first performed MAGMA gene-property analysis on the FUMA platform[64] to identify tissue specificity based on the gene expression profiles of 54 tissues from the GTEx v.8 project[65]. The gene-property analysis was based on the regression model $Z \sim \beta_0 + E_t \beta_E + A\beta_A + B\beta_B + \epsilon$, where Z was the gene-based Z-score converted from the gene-based $P$-value, $B$ was the matrix of several technical confounders included by default. $E_t$ was the gene expression value of a testing tissue type, $A$ was the average expression across tissue types in a dataset, and $\epsilon$ was the random error. We performed a one-sided test ($\beta_E > 0$) which was essentially testing the positive relationship between tissue specificity and genetic associations.

Then, the stratified LD score regression (LDSC)[66] was applied to test if KSD heritability was enriched in 220 cell type-specific genomic functional regions. The stratified LDSC model was $E(\chi_i^2) = 1 + N\alpha + N\sum_k \tau_k l(i,k)$, where $\chi_i^2$ was the GWAS summary statistic for SNP i, N was the GWAS sample size, $\alpha$ was a constant that reflects population structure and other sources of confounding, $l(i,k)$ was the LD score of SNP i in annotation k, and $\tau_k$ is the regression coefficient of cell type-specific annotation LD score. We included 53 functional baseline annotations that were not specific to any cell type (e.g., coding, 3′ UTR, 5′ UTR, promoter and intron)[66,67]. The inclusion of annotations common to all cell types can help remove potential confounding factors and enhance the cell type-specific signals. The baseline LD score and 220 cell type-specific annotation LD score have

 

been calculated based on the genotypes of European in the 1000 Genomes Project[67] and downloaded from https://alkesgroup.broadinstitute.org/LDSCORE/. In the stratified LDSC model, the coefficient of cell type-specific annotation LD score quantified the importance of annotation. Thus, we computed $P$ value that tested whether the coefficient of cell type-specific annotation LD score was positive ($\tau_k > 0$).

We further explored the distribution of KSD associated variants in different cell type-specific accessible regions identified by snATAC-seq in two human kidney samples (gene expression omnibus accession number GSE172008)[68]. The Chi-square test for a four-fold table was used for the enrichment test and the enrichment was defined as the odds ratio.

### Gene mapping and prioritization
We applied three approaches to map and prioritize the GWAS associated variants to the protein-coding genes, including kidney-specific eQTL mapping, kidney-specific meQTL/eQTM mapping and kidney-specific TWAS and colocalization.

**Kidney-specific eQTL mapping.** Significant kidney-specific eQTLs in the tubule (n = 356 for dataset of Sheng et al., 121 for dataset of Qiu et al., false discovery rate (FDR) < 0.05), glomeruli (n = 303 for dataset of Sheng et al., 119 for dataset of Qiu et al., FDR < 0.05)[68,69] and human kidney meta-analysis (n = 696, FDR < 0.01)[62] were downloaded from Susztak lab (http://www.susztaklab.com/). We annotated the GWAS associated variants to the genes based on kidney-specific eQTL.

**Kidney-specific meQTL/eQTM mapping.** Significant kidney-specific meQTLs (n = 443, FDR < 0.05) and eQTM (n = 415, FDR < 0.05)[62] were also downloaded from Susztak lab (http://www.susztaklab.com/). We first annotated the significant meQTL with KSD associated variants which were mapped to the probe whose methylation level was affected by the variants. Then we further used eQTM data to map the gene that may be regulated by this methylation probe. The integration of meQTL and eQTM provided a new way to prioritize target genes for GWAS signals[62].

**Kidne-specific TWAS and colocalization.** We performed transcriptome-wide association study (TWAS) analysis with the FUSION[70] pipelines using the kidney-specific eQTL information from the GTEx v.8 project (kidney cortex, n = 73)[65]. Due to the cis-genetic effects on gene expression being highly consistent between disease cases and controls in the datasets to derive the TWAS weight[71], we also used the gene expression profiles in kidney renal clear cell carcinoma (KIRC, n = 506) and kidney renal papillary cell carcinoma (KIRP, n = 283) from the cancer genome atlas (TCGA) project. The TWAS weights were downloaded from http://gusevlab.org/projects/fusion/. The numbers of protein-coding genes finally tested were 14,476 for GTEx kidney cortex, 2,622 for KIRC and 1946 for KIRP. The Bonferroni-corrected $P$ value thresholds were applied for each dataset separately. We further performed Bayesian colocalization analysis to investigate whether the causal variants of KSD and gene expression were shared. Bayesian colocalization method[72] was also implemented in the FUSION[70] pipelines. The posterior probability that a variant associated with KSD and kidney-specific gene expression was estimated. Posterior probability >0.8 was considered as the evidence for colocalization.

### Pathway enrichment analysis with the prioritized genes
Associations of GO biological processes (BP) pathways with the prioritized genes were investigated using the g:GOSt tool implemented in the web server, g:Profiler, which is updated approximately in every three months and follows quarterly releases of Ensembl database[73]. The prioritized genes were assumed to be enriched in a pathway if the $P$-value was less than 0.05 after multiple testing correction. We excluded the prioritized genes in the major histocompatibility complex (MHC) region and used the autosomal protein-coding genes as the background.

### Drug-gene interaction analysis
We first annotated the prioritized genes with known drug-gene interactions and potential druggability by searching in the drug-gene interaction database (DGIdb, www.dgidb.org). Drugs provided by this database are defined by ChEMBL, which is a manually curated database of bioactive molecules with drug-like properties. Next, we identified the medicines that contain the above drug ingredients for KSD by searching through the WebMD website (https://www.webmd.com/drugs).

### Phenome-wide genetic correlation
We downloaded the GWAS summary statistics of 2202 diseases from FinnGen (Beta 8) and excluded KSD. The genetic correlations with KSD were estimated with cross-trait LDSC[74] which applied a weighted linear model by regressing the product of Z-statistics of pairwise diseases on the LD scores of SNPs across the whole genome. The regression slope provides an unbiased genetic correlation estimate for pairwise diseases even when samples overlap in the two studies. For the genetic correlation and the cell type-specific analyses, we only used the GWAS summary statistics of HapMap3 SNPs and further removed the SNPs in the MHC region. Finally, 1781 pairs of genetic correlations were successfully estimated and the Bonferroni-correlated $P$ value threshold of $2.81 \times 10^{-5}$ was used to define the significant genetic correlations.

### Reporting summary
Further information on research design is available in the Nature Portfolio Reporting Summary linked to this article.

## Data availability
Full UKB data are available by direct application to UKB. GWAS summary statistics of FinnGen are available at https://r8.finngen.fi/. The full GWAS summary statistics of KSD is public available at https://zenodo.org/records/10060271. Further inquiries can be directed to the corresponding author X.H.

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

## Acknowledgements

This work was supported by National Natural Science Foundation of China (Award number: 82003561, 82325044, 82021005), Natural Science Fund for Distinguished Young Scholars of Hubei Province (2022CFA046), Fundamental Research Funds for the Central Universities (2023BR030), and Fundamental Research Funds for School of Public Health, Tongji Medical College, Huazhong University of Science and Technology (2022gwzz01). We thank participants and investigators who contributed to the GWAS summary statistics and omics data included in our analyses.

## Author contributions

Study design: X.H. and C.W.; data collection, analysis and interpretation: Z.S., N.Z., M.J., S.L., X.C., Y.G., and X.H.; manuscript writing: N.Z., M.J., and X.H.; All authors participated in the preparation of manuscript by reading and commenting on the draft prior to submission.

## Competing interests

The authors declare no competing interests.
