## [Peer Review File · Nature Communications]

Integrative genome-wide analyses identify novel loci associated with kidney stones and provide insights into its genetic architectureREVIEWER COMMENTS

Reviewer #1 (Remarks to the Author):

Hao et al report their findings in the largest GWAS meta-analysis on kidney stone disease to date. They achieved this large sample size by compiling data from the UKB and FinnGen (> 17,900 cases). By doing so they identified 44 susceptibility loci, including 28 novel loci. The authors further prioritized 223 genes by kidney specific OMIC datasets (e.g. eQTL). However, the manuscript remains very superficial in terms of data interpretation and discussion; as a result, the authors conclude with very obvious findings and generic statements such as "the importance of ion homeostasis in stones formation". In order to improve the article, the following points need to be addressed:

Major:

- Introduction: "coffee and caffeine consumption" is thought to be protective in kidney stone formation – please revise as it currently evokes the opposite.
- Results: "Five previously identified loci (i.e., 2p23.3, 11q23.1, 15q21.3, 16p12.2 and 19p13.12) ..." – please name the primary candidate gene for each locus to increase readability and transparency.
- Results: "In addition, another drug hydrochlorothiazide can prevent kidney problems by making patients more urinate and reducing extra fluid in the body (edema) caused by conditions such as heart failure, liver disease, or kidney disease." - please revise, this is not the commonly postulated mechanism why HCT may be beneficial in kidney stone metaphylaxis. However, very recently the beneficial effect of HCT has become controversial anyway (PMID: 36856614).
- Figure 2: for more clarity, each locus should be labelled with the primary candidate gene in the Manhattan plot.
- Results: Why did the authors not investigate additional biochemical traits from UKB/FinnGen, such as hypercalcemia/hypophosphatemia/hypercalciuria in their dataset to strengthen obtained associations (PMID: 35197637)?
- Results: Why did the authors not try to impute protein coding variants that may directly be causative for observed associations (PMID: 35197637)?
- Results: Basic confirmatory in vitro data would be highly appreciated to support significance of some of the newly proposed candidate genes (as previously done in PMID: 31729369)
- Discussion: The putative role of the novel candidate genes in formation of kidney stones is not sufficiently discussed.
- General remark: The manuscript would benefit from language editing by a native speaker (exclusively written in past tense etc.)

Minor:

- Introduction: "... monogenic cause of kidney stones has been identified in up to 30% of children and 10% of adults ..." – please cite original articles not a single review to reference this phrase.
- Introduction: "... calcium-sensing receptor signaling pathway, ions, protons and amino acids ..." – please revise: what do the authors mean by "ions" and "protons"? A proton can be an ion too.

Reviewer #2 (Remarks to the Author):

This study of the genetic architecture of kidney stone disease makes use of publicly available data in combination with data from the UK Biobank. The authors conduct a GWAS of kidney stone disease in the UK Biobank and undertake a meta-analysis of these data with summary statistics released from FinnGen. They use summary statistics that have been released from a Biobank Japan GWAS to look for replication of novel kidney stone associated loci.

Subsequently they explore possible causal pathways using eQTL, meQTL, eQTM and TWAS data focusing on the kidney on the basis of tissue specificity analyses undertaken in MAGMA and availability

of these datasets from the kidney. With this approach they identify 223 possible causal genes and match these genes with data from the DGIdb database to look for possible drug modifiers. Finally, using FinnGen summary statistics they look for genetic correlations of kidney stone disease with other disorders.

This manuscript provides a comprehensive GWAS of kidney stones with novel loci identified. Potential causal genes are identified, but no attempt is made to validate or describe the importance or mechanistic role of these genes. Drugs that may interact with these genes are identified using DGIb but no attempts are made to ascertain whether these drugs would ameliorate or potentiate a kidney stone phenotype. Comment is made on genetic correlation with other disease phenotypes in FinnGen but there is no validation in other data sets that these correlations hold.

In summary, this study provides data relating to the genetic architecture of kidney stone disease but with little validation and without a narrative that leads to clear conclusions about the pathophysiology of this disorder as a whole.

Some other more specific comments:

The introduction notes: 'The formation of kidney stones has a multifactorial aetiology involving both genetic and environmental factors, such as coffee and caffeine consumption, calcium and vitamin D supplementation'

There are many more clear cut environmental influences that lead to kidney stone formation (for example dehydration) - the examples used seem odd choices to cite. Calcium supplementation can be beneficial in a subset of patients, the role of vitamin D supplementation is controversial, and caffeine appears to be protective rather than a risk factor as is implied by the text.

Patients are ascertained from the UK Biobank using only N20 coding from ICD10, using ICD 9 as well and extending to N23 and OPCS codes would improve ascertainment of stone cases.

A conditional analysis, rather than a clumping approach would improve definition of independent loci.

The threshold used to ascertain replication in the Biobank Japan I would not consider to be sufficiently stringent, and would suggest 5×10^{-5} .

Is there any evidence that the genes that are thought to be causal are expressed at protein level in the kidney? If so, does the tubular expression pattern match with the open chromatin patterns in the snATAC data?

The authors state: ' In addition, another drug hydrochlorothiazide can prevent kidney problems by making patients more urine and reducing extra fluid in the body (edema) caused by conditions such as heart failure, liver disease, or kidney disease.' Thiazides reduce urinary calcium levels and are thought to prevent kidney stones in this way. However, a recent RCT suggests they may not be effective: Dhayat NEJM, Hydrochlorothiazide and Prevention of Kidney-Stone Recurrence.

The role of vitamin D is discussed in the conclusion, this may be the mechanism but seems a leap from the data presented. It would be possible to look for correlations of genetic variants with vitamin D concentrations at GWAS to ascertain whether there are potential influences on vitamin D absorption

REVIEWER COMMENTS

Reviewer #1 (Remarks to the Author):

Hao et al report their findings in the largest GWAS meta-analysis on kidney stone disease to date. They achieved this large sample size by compiling data from the UKB and FinnGen (> 17,900 cases). By doing so they identified 44 susceptibility loci, including 28 novel loci. The authors further prioritized 223 genes by kidney specific OMIC datasets (e.g. eQTL). However, the manuscript remains very superficial in terms of data interpretation and discussion; as a result, the authors conclude with very obvious findings and generic statements such as "the importance of ion homeostasis in stones formation". In order to improve the article, the following points need to be addressed:

Response: We thank the reviewer for the comments and suggestions for our work. In our revised manuscript, we extended our interpretation and discussion about genes and potential mechanism (*Lines 250-257, 269-324*). Our tissue and cell type-specific enrichment analysis indicated the formation of kidney stones was influenced not only by the metabolism and absorption of vitamin D and calcium in the liver and intestine, but also by the calcium reabsorption in the kidney. Our findings not only underscored the importance of ion imbalance in kidney stones, but also indicated that multiple complex biological processes participate in the pathophysiology of kidney stones. For the potential mechanism of the genes, here are some examples that we added in our revised manuscript.

1. *CYP24A1* had the highest expression in kidney based on GTEx profiles, and the risk allele of rs17216707 (T) for kidney stones at *CYP24A1* was significantly associated with higher calcium, phosphate and vitamin D concentration. These evidences suggested a potential mechanism that the T allele at rs17216707 could increase the risk of kidney stones by increasing the vitamin D concentration, then increasing the reabsorption of calcium and phosphate in the kidney.

2. At the novel locus 1q32.1, *PM20D1* involved in the lipid metabolic process and regulating energy homeostasis, which may influence the transmembrane transport. In addition, serum *PM20D1* level was significantly elevated in overweight and obese individuals, who had a higher risk to develop kidney stones. *SLC41A1* was essential for

magnesium homeostasis in vivo. Individuals with kidney stones tended to have lower magnesium in blood compared to people without kidney stones, potentially because the magnesium can inhibit calcium oxalate kidney stones by binding with oxalate, thus making oxalate less likely to bind with calcium to form kidney stones. In addition, *SLC45A3* and *SLC26A9* were involved in the biological process of ion transport and regulation of pH, which was very important for the kidney stone formation.

3. At the novel locus 17q21.31, *LRRC37A2* and *LRRC37A* were predicted to be the integral component of membrane and involved in protein binding, which may participate in transmembrane transport. *WNT3* has been shown to play a leading role during bladder development and was implicated in the pathogenesis of the group of conditions called bladder-exstrophy-epispadias complex, which has a profound impact on renal functions. *PLEKHMI* was involved in biological process of metal ion binding and associated with osteopetrosis, which had a very high prevalence (about 23.6%) in patients with kidney stones. Our results suggested a potential shared regulatory mechanism of *PLEKHMI* for osteopetrosis and kidney stones.

4. Based on the position mapping of the novel loci, we identified multiple genes involved in ion transportation. For example, we found *SLC30A10* in 1q41 encodes manganese transporter, which may regulate manganese concentration then can impact the formation of calcium oxalate kidney stones. Previous studies observed the manganese concentration in urine was lower in patients with kidney stones and the higher manganese intake was associated with a lower risk of kidney stones. Calcium (Ca^{2+}) plays a key role in stone formation, although many candidate gene association studies were focused on the epithelial Ca^{2+} channel transient receptor potential vanilloid (*TRPV5*), no genome-wide significant signals ($P < 5 \times 10^{-8}$) associated with kidney stones were identified at this locus probably due to the small sample size. In this study, we identified a novel signal located 46 bp downstream of the *TRPV5* showing an association with kidney stones (rs4252512, MAF=0.015, OR=0.78, $P=1.88 \times 10^{-8}$), which may help understand the role of *TRPV5* in the pathology of kidney stones. Notably, we identified a rare missense variant, rs34819316 (MAF=0.005, OR=0.54,

$P=4.58 \times 10^{-8}$) in *SLC12A1*, significantly associated with kidney stones (Supplementary Table 6). *SLC12A1* encodes a kidney-specific sodium-potassium-chloride cotransporter and functions in concentrating urine and NaCl reabsorption⁴⁶. Mutations in *SLC12A1* could cause Bartter syndrome, which is correlated with hypercalciuria.

More interpretation and discussion about genes and potential mechanism (*Lines 250-257, 269-324*) were provided in our revised manuscript. We thank the reviewer again for suggestions and believe that the manuscript is improved as a result.

Major:

- Introduction: “coffee and caffeine consumption” is thought to be protective in kidney stone formation – please revise as it currently evokes the opposite.

Response: We revised it and used more common risk factors, such as obesity, dehydration and diet with high sodium.

- Results: “Five previously identified loci (i.e., 2p23.3, 11q23.1, 15q21.3, 16p12.2 and 19p13.12) ...”, please name the primary candidate gene for each locus to increase readability and transparency.

Response: Thanks for your suggestion, the primary candidate gene for each locus was provided in our revised manuscript.

- Results: “In addition, another drug hydrochlorothiazide can prevent kidney problems by making patients more urine and reducing extra fluid in the body (edema) caused by conditions such as heart failure, liver disease, or kidney disease.” - please revise, this is not the commonly postulated mechanism why HCT may be beneficial in kidney stone metaphylaxis. However, very recently the beneficial effect of HCT has become controversial anyway (PMID: 36856614).

Response: Thanks for the reminder. Thiazide diuretic agents are widely used for prevention of the recurrence of kidney stones. Recently, Dhayat NA, et al utilized a double-blind trial and concluded that HCT did not play a role in preventing the recurrence of renal stones (*N Engl J Med.* 2023;388(9):781-791). However, these

patients were only followed for a median of 2.9 years, whether the conclusion can be applied to long-acting thiazide diuretic agents remains to be determined. In addition, they also found that in patients with radiological recurrence (secondary endpoint), the recurrence rate was lower in the hydrochlorothiazide group, and the effect was more pronounced at higher doses. In our revised manuscript, we mentioned the latest paper and extended our discussion.

- Figure 2: for more clarity, each locus should be labelled with the primary candidate gene in the Manhattan plot.

Response: Thanks for your suggestion, the primary candidate genes have been labelled in the Manhattan plot in our revised manuscript.

- Results: Why did the authors not investigate additional biochemical traits from UKB/FinnGen, such as hypercalcemia/hypophosphatemia/hypercalciuria in their dataset to strengthen obtained associations (PMID: 35197637)?

Response: Thanks for your suggestion. We investigated associations between the lead variants at the susceptibility loci with calcium, phosphate or 25 hydroxyvitamin D concentration in blood. We found that the lead variants at 12 loci were associated with calcium, phosphate or 25 hydroxyvitamin D concentration in blood (Supplementary Tables 3 and 4). Notably, the risk allele of rs17216707 (T) for kidney stones at the known locus *CYP24A1* was significantly associated with higher calcium (beta=0.059, $P=8.12 \times 10^{-76}$), phosphate (beta=0.025, $P=6.65 \times 10^{-15}$) and 25 hydroxyvitamin D concentration (beta=0.038, $P=3.47 \times 10^{-46}$). We have added these results (Supplementary Tables 3-4) and extended our discussion in our revised manuscript.

- Results: Why did the authors not try to impute protein coding variants that may directly be causative for observed associations (PMID: 35197637)?

Response: Thanks for your suggestion. We tried to investigate the protein coding variants, and found that 48 protein coding variants, including 26 missense variants, showed significant association with kidney stones. Specifically, these missense variants

located in 20 unique genes. We added these results in the manuscript and Supplementary Table 6. Notably, we identified a rare missense variant, rs34819316 (MAF=0.005, OR=0.54, $P=4.58\times 10^{-8}$) in *SLC12A1*, significantly associated with kidney stones. *SLC12A1* encodes a kidney-specific sodium-potassium-chloride cotransporter and functions in concentrating urine and NaCl reabsorption. Mutations in *SLC12A1* could cause Bartter syndrome, which is correlated with hypercalciuria.

- Results: Basic confirmatory *in vitro* data would be highly appreciated to support significance of some of the newly proposed candidate genes (as previously done in PMID: 31729369)

Response: Thanks for your suggestion. We are sorry for not conducting *in vitro* experiment, which is very difficult for our bioinformatics and statistical genomics lab. However, most of the loci can be validated in other independent sample from Japan. We additionally conducted the validation study in Michigan Genomics Initiative project. The available lead variants at 27 novel loci showed similar effect sizes with the same sign direction (Supplementary Table 5 and Supplementary Fig. 2). Specifically, the prioritized genes based on GWAS summary statistics and multi-omics data included many known susceptibility genes, which highlighted our approach was effective and useful. In addition, the newly proposed candidate genes were supported by eQTL and meQTL from multiple sources and identified by multiple approaches. For example, *PM20D1* was identified by three approaches in seven eQTL/meQTL datasets. We have extended our discussion for the novel genes and admitted the limitation in our study, and also suggested functional validations in the future, especially for the new genes supported by multiple sources. We are willing to search for cooperation for *in vitro* experiment if the reviewer and/or editors have a strong preference.

- Discussion: The putative role of the novel candidate genes in formation of kidney stones is not sufficiently discussed.

Response: Thanks for your suggestion. We have extended our discussion about the novel candidate genes in the novel loci or prioritized by kidney specific omics data.

- General remark: The manuscript would benefit from language editing by a native speaker (exclusively written in past tense etc.)

Response: Thanks for your suggestion. We have carefully revised the manuscript.

Minor:

- Introduction: "... monogenic cause of kidney stones has been identified in up to 30% of children and 10% of adults ..." – please cite original articles not a single review to reference this phrase.

Response: Thanks for your suggestion. The original articles (PMID: 25986801, PMID: 26787776, PMID: 28893421) have been cited in our revised manuscript.

- Introduction: "... calcium-sensing receptor signaling pathway, ions, protons and amino acids ..." – please revise: what do the authors mean by "ions" and "protons"? A proton can be an ion too.

Response: Thanks for your suggestion. We deleted protons and only used ions in our revised manuscript.

Reviewer #2 (Remarks to the Author):

This study of the genetic architecture of kidney stone disease makes use of publicly available data in combination with data from the UK Biobank. The authors conduct a GWAS of kidney stone disease in the UK Biobank and undertake a meta-analysis of these data with summary statistics released from FinnGen. They use summary statistics that have been released from a Biobank Japan GWAS to look for replication of novel kidney stone associated loci. Subsequently they explore possible causal pathways using eQTL, meQTL, eQTM and TWAS data focusing on the kidney on the basis of tissue specificity analyses undertaken in MAGMA and availability of these datasets from the kidney. With this approach they identify 223 possible causal genes and match these genes with data from the DGIdb database to look for possible drug modifiers. Finally, using FinnGen summary statistics they look for genetic correlations of kidney stone disease with other disorders.

Response: We thank the reviewer for the accurate summary for our work.

This manuscript provides a comprehensive GWAS of kidney stones with novel loci identified. Potential causal genes are identified, but no attempt is made to validate or describe the importance or mechanistic role of these genes. Drugs that may interact with these genes are identified using DGIdb but no attempts are made to ascertain whether these drugs would ameliorate or potentiate a kidney stone phenotype. Comment is made on genetic correlation with other disease phenotypes in FinnGen but there is no validation in other data sets that these correlations hold. In summary, this study provides data relating to the genetic architecture of kidney stone disease but with little validation and without a narrative that leads to clear conclusions about the pathophysiology of this disorder as a whole.

Response: Thanks for your comments. We have added many new analyses and especially extended the results and discussions which were highlighted in our revised manuscript. For example,

1. For the novel loci, we additionally validated them in Michigan Genomics Initiative project, the available lead variants at 27 novel loci showed similar effect size

with the same sign direction (Supplementary Figure 2).

2. To describe the importance or mechanistic role of the susceptibility loci, we investigated associations between the lead variants at the susceptibility loci with calcium, phosphate or 25 hydroxyvitamin D concentration in blood (Supplementary Tables 3 and 4). The prioritized genes were supported by eQTL and meQTL from multiple sources and identified by multiple approaches. In addition, more than half of the prioritized genes were expressed at protein level detected by microarray-based immunohistochemistry in kidney (Supplementary Table 12). Moreover, the prioritized genes included many known susceptibility genes, which highlighted our approach was effective and useful. Although we did not conduct the in vitro experiment to validate the prioritized genes, we extended the description and discussion about the potential roles of novel genes in formation of kidney stones.

3. We extended the description and discussion about the drugs. We understand that our inability to perform experiments to test the efficacy of the identified drugs in the context of kidney stone formation may be a limitation of our study. However, our study aimed to provide a preliminary understanding of potential drug-gene interactions that could be further explored in future research. We hope that the findings of our study can contribute to a broader body of knowledge about kidney stone formation and inform future studies in this area.

4. For genetic correlations, we found that kidney stone disease has almost perfect genetic correlations with calculus of lower urinary tract ($r_g=1.0627$, another stone disease in the genitourinary system) and calculus of kidney and ureter ($r_g=1.0495$, part of our kidney stones GWAS-meta analysis) in FinnGen (Supplementary Table 15), which could be positive controls and highlighted genetic correlation analyses were useful and accurate. Actually, most phenome-wide genetic correlation studies were not validated in another dataset (*Nature genetics* 2019; 51(9): 1339-48., *Nature genetics* 2021; 53(9): 1276-82., *Nature genetics* 2021; 53(6): 817-29., *Nature Communications* 2023; 14(1): 157., *Nature genetics* 2023; 55(2): 198-208.), which may mainly due to the difficulty to collecting thousands of GWASs for the same phenotypes in two large datasets or biobanks. In addition, LDSC was widely used and popular for its robust and

accurate estimation of genetic correlation.

We thank the reviewer again for suggestions and believe that the manuscript is improved as a result.

Some other more specific comments:

The introduction notes: 'The formation of kidney stones has a multifactorial aetiology involving both genetic and environmental factors, such as coffee and caffeine consumption, calcium and vitamin D supplementation'. There are many more clear cut environmental influences that lead to kidney stone formation (for example dehydration) - the examples used seem odd choices to cite. Calcium supplementation can be beneficial in a subset of patients, the role of vitamin D supplementation is controversial, and caffeine appears to be protective rather than a risk factor as is implied by the text.

Response: Thanks for your comments. We have revised it and used more common risk factors, such as obesity, dehydration and diet with high sodium

Patients are ascertained from the UK Biobank using only N20 coding from ICD10, using ICD 9 as well and extending to N23 and OPCS codes would improve ascertainment of stone cases.

Response: Thanks for your suggestion. Actually, we derived kidney stones phenotype using any code mapped to 3-character ICD-10 (category ID 1712), which was generated by mapping ICD-10 code in the death register records, read code information in the primary care data, ICD-9 and ICD-10 codes in the hospital inpatient records (*e.g.*, diagnosis or operation code), and self-reported medical conditions reported at the baseline or subsequent visit to UK Biobank assessment center. The ascertainment of stone cases should be comprehensive and accurate. We have revised the method parts about the case definition and source.

As suggested by the reviewer, we extended the case definition by including N23 (unspecified renal colic). After meta-analysis with FinnGen study, 38 independent and significant loci, including 22 novel loci were identified. However, all these novel loci have been identified in our previous meta-analysis which not including N23 cases. So

we used our previous GWAS meta-analysis which identified more novel loci and had more accurate case definition as our main results for the following post-GWAS integrative analysis. We also provided the results including N23 as cases in the Supplementary Figure 1

A conditional analysis, rather than a clumping approach would improve definition of independent loci.

Response: Thanks for your suggestions. Here we applied another commonly used approach (*Nature genetics* 2022; 54(8): 1125-32.; *Nature genetics* 2022; 54(7): 950-62., *Nature genetics* 2019; 51(3): 394-403., *Nature genetics* 2018; 50(7): 920-7.) to define the significant and independent loci by choosing the lead and independent variants firstly, then merging the nearby (<1000kb) lead variants into the same locus. Each locus is represented by the most significant (top) SNP. We agree with the reviewer, and acknowledge that some loci in our study may have more than one independent “signals” or “variants” (here we don’t use “loci” to distinguish the definition). By applying such an approach in our study, a larger genomic region was defined as a susceptibility locus. Therefore, it was relatively accurate and conservative to define where the locus was previously reported or novel.

As the reviewer suggested, we additionally conducted the stepwise conditional analysis in each locus using GCTA-COJO. There were 56 independent signals in 44 loci. The number of independent signals in each locus ranged from one to six. The information of independent signals was also provided in the Supplementary Table 1.

The threshold used to ascertain replication in the Biobank Japan I would not consider to be sufficiently stringent, and would suggest 5×10^{-5} .

Response: Thanks for your comments. For GWAS replication study, the Bonferroni corrected P value threshold ($0.05/\text{number of loci need to be validated}$) is often used and conservative. In addition, we have provided the detailed p values of variants in the BBJ in Table 1 for other researchers to judge. As the reviewer suggested, we also stated that eight loci were replicated when using 5×10^{-5} threshold in our revised manuscript.

Is there any evidence that the genes that are thought to be causal are expressed at protein level in the kidney? If so, does the tubular expression pattern match with the open chromatin patterns in the snATAC data?

Response: Thanks for your suggestion. Unfortunately, the single cell proteomics data is not available in kidney currently. We cannot compare the tubular protein expression pattern with the open chromatin patterns in the snATAC data. However, the tubular gene expression pattern matched with the open chromatin patterns in the snATAC data in the previous study (*Nature Communications* 2021; 12(1): 2277.).

To provide more evidence for the 223 prioritized genes, we checked their protein level in bulk kidney tissue at the human protein atlas (<https://www.proteinatlas.org/>, Science 2015; 347(6220): 1260419). We found that 126 genes were expressed at protein level detected by microarray-based immunohistochemistry in kidney, including proximal tubules, distal tubules, collecting ducts, cells in tubules, cells in glomeruli and bowman's capsule. The detailed information has been added in our revised manuscript and Supplementary Table 12.

The authors state: ' In addition, another drug hydrochlorothiazide can prevent kidney problems by making patients more urine and reducing extra fluid in the body (edema) caused by conditions such as heart failure, liver disease, or kidney disease.' Thiazides reduce urinary calcium levels and are thought to prevent kidney stones in this way. However, a recent RCT suggests they may not be effective: Dhayat NEJM, Hydrochlorothiazide and Prevention of Kidney-Stone Recurrence.

Response: Thanks for pointing this out. We have revised the sentences as “... it decreases the amount of calcium excreted by kidney in the urine and thus decreases the amount of calcium in urine to form stones”. As our response to the 1st reviewer for latest paper, thiazide diuretic agents are widely used for prevention of the recurrence of kidney stones. Recently, Dhayat NA, et al utilized a double-blind trial and concluded that HCT did not play a role in preventing the recurrence of renal stones (*N Engl J Med.* 2023;388(9):781-791). However, these patients were only followed for a median of 2.9

years, whether the conclusion can be applied to long-acting thiazide diuretic agents remains to be determined. In addition, they also found that in patients with radiological recurrence (secondary endpoint), the recurrence rate was lower in the hydrochlorothiazide group, and the effect was more pronounced at higher doses. In our revised manuscript, we mentioned the latest paper and extended our discussion.

The role of vitamin D is discussed in the conclusion; this may be the mechanism but seems a leap from the data presented. It would be possible to look for correlations of genetic variants with vitamin D concentrations at GWAS to ascertain whether there are potential influences on vitamin D absorption

Response: Thanks for your suggestion. This is a good question. The previous studies have explored the correlations of kidney stones associated genetic variants with vitamin D concentrations (Table 2 in *Nature Communications* 2019; 10(1): 5175.; Table 2 in *Nature Communications* 2015; 6(1): 7975.), however no correlations were found. In our study, we found a positive but not significant genetic correlation ($r_g=0.3297$, $se=0.2174$, $P=0.1295$) between Vitamin D deficiency and kidney stones in our phenome-wide genetic correlation analysis. We further explored the genetic correlation between vitamin D concentration (*Nature Communications* 2020; 11(1): 1647.) with kidney stones, and found a negative but not significant genetic correlations ($r_g=-0.052$, $se=0.0303$, $P=0.0862$). The global genetic correlation analysis indicated that vitamin D concentration may not share lots of common genetic bases with kidney stones.

For single genetic variant level, we investigated associations between the lead variants at the susceptibility loci with calcium, phosphate or 25 hydroxyvitamin D concentration in blood (see our response to the 1st reviewer). We found two independent and significant lead variants for kidney stones were associated with vitamin D concentration (Supplementary Table 4), including rs17216707 at the known loci covering *CYP24A1* which was predicted to affect vitamin D metabolism. In addition, the risk allele of rs17216707 (T) for kidney stones was associated with higher ($\beta>0$) vitamin D concentration, which could increase the absorption of calcium. Moreover, *CYP24A1* had the highest expression in kidney based on GTEx profiles (Supplementary

Figure 3). These results highlighted our cell type-specific enrichment analysis, which suggested that the candidate genes may function in a cell type-specific manner. In summary, we added more results and discussion in the revised manuscript.

REVIEWER COMMENTS

Reviewer #1 (Remarks to the Author):

The revised manuscript is overall improved, notably the Discussion. Although genetic validation has now been provided (Michigan Genomics Initiative) independent validation from in vitro data is still missing, which remains the greatest limitation.

Beyond that, there are some remaining aspects that should be addressed, notably in terms of the novel candidate genes:

Discussion:

- Please add OMIM-reference number for all newly identified candidate genes and briefly mention the phenotype of the respective Mendelian disease, if there is one.
- PM20D1 (line 271-273): Please rephrase this sentence as it sounds like the study that found an association of PM20D1 levels with obesity also found an association with kidney stone disease (KSD), which is not the case. I recommend to make two separate sentences out of this statement for clarification.
- SLC41A1: to be critically discussed that the only Mendelian association of the gene refers to a Nephronophthisis-like phenotype but no kidney stone disease (KSD) or nephrocalcinosis (MIM# 619468; PMID: 23661805).
- CLDN10: please discuss that patients with the corresponding Mendelian condition (HELIX-Syndrome, MIM# 617671, PMID: 28771254) do not show KSD or nephrocalcinosis.
- TRPV5: this is a long-standing candidate gene for hypercalciuria and KSD, as ko mice show severe hypercalciuria (PMID: 14679186) - please cite this reference for completeness and acknowledgement.
- SLC12A1: interesting locus as this is the Bartter Type 1 gene - please add the MIM-number (#601678) as in all Mendelian conditions. Loss-of-function of this gene is not only correlated with hypercalciuria but a very severe (mostly syndromic) disorder associated with nephrocalcinosis and kidney failure - gene discovery publication may be relevant to cite: PMID: 8640224
- SLC30A10: interesting candidate as well - please add MIM# 613280 and reference for Mendelian hypermanganesemia PMID: 22341972.

Of note: the manuscript still needs final editing by a native speaker to enhance both readability and comprehensibility.

Reviewer #2 (Remarks to the Author):

The revisions have improved the manuscript, however there remains a superficiality to the links between identified loci of interest and their function, and it is of some concern that the genes noted to be of interest in the discussion are often only supported by one of two pieces of evidence and potential pathophysiology may not have been fully explored. For example, PLEKHM1 is highlighted due to its links with osteopetrosis and an analogy drawn with altered BMD in stone formers. However, the study cited concerns osteoporosis in stone formers, which could be considered the opposite phenotype; I think there has been confusion regarding osteopetrosis and osteoporosis.

The authors discuss how CYP24A1 may have relevance with regard to vitamin D concentrations, the GWAS summary statistics they have looked to for correlations are based on 25-OH data, CYP24A1 inactivates 1,25-OH vit D, a few steps away from 25-OH, a mechanism which is not noted or discussed.

PM20D1 is highlighted as it has been found to be have increased concentrations in overweight individuals in one study. It is well known that adiposity increases risk of kidney stones. However, it seems a leap to suggest that PM20D1 is responsible for this link, and PM20D1 does not appear to be renally expressed.

SLC12A1 has in rare cases been linked to Bartters, but isn't the more likely explanation that there is

some effect on water reabsorption to affect renal tubular salt concentrations?

I apologise if I missed part of the file download, but I was unable to find the figure legends which has hampered my ability to comment fully on these. I may have misunderstood but in figure 3B your text states:

'Among 220 cell type specific chromatin modification marks, six cell types, including three kidney cell types, were relevant to kidney stone disease'

Isn't it the case that there are six cell specific chromatin modification marks that have been identified rather than six cell types? And within this 3 kidney specific markers?

Within the pathway enrichment analysis the most significantly enriched gene set was "appendage morphogenesis". What is the explanation for this? Does this maybe suggest that gene prioritisation hasn't resulted in reliable results as I can't see a link between limb development and stones. In addition I think GO:0072502 is now obsolete?

With regard to drug-gene interaction pairs, how have the 38 drugs for renal calculi prevention been defined? I am not clear that there are 38 different drugs available to prevent stones? Is there some overlap between these drugs?

'In addition, we found a positive but insignificant genetic correlation ($r_g=0.33$, $P=0.13$) between kidney stones and vitamin deficiency'. This seems an odd way to report to me, isn't the finding here that there is no association between vit D deficiency and stones?

The article states that stones can block the ureter to cause infection. It would be my understanding that a urinary infection co-exists with a stone, the blockage means the infection can not adequately clear, it is not the case that the stone causes the infection.

The snATAC seq presented I think is from mouse tissue rather than human, this should be highlighted. Human scATACseq data is publicly available and would be preferable to use.

Reviewer #1 (Remarks to the Author):

The revised manuscript is overall improved, notably the Discussion. Although genetic validation has now been provided (Michigan Genomics Initiative) independent validation from in vitro data is still missing, which remains the greatest limitation.

Response: We thank the reviewer for acknowledging our effort to improve the manuscript. We agree with reviewer that the missing independent validation by vitro experiment was a limitation for our study as we mentioned in the discussion part, although we provided a comprehensive atlas of susceptibility genes for kidney stone disease which can guide the follow-up studies in physiology, basic science, and clinical medicine. We have carefully revised the manuscript based on your comments and highlighted the changes in our new manuscript. The remaining comments are addressed below.

Beyond that, there are some remaining aspects that should be addressed, notably in terms of the novel candidate genes: Discussion:

- Please add OMIM-reference number for all newly identified candidate genes and briefly mention the phenotype of the respective Mendelian disease, if there is one.

Response: Thanks for your suggestion. We have added the OMIM number and Mendelian diseases for the genes at the novel loci (Supplementary Table 5) and prioritized genes (Supplementary Table 8). For example, *WNT3* also showed a Mendelian association with tetra-amelia syndrome 1 (#273395), and renal and adrenal agenesis have been observed in tetra-amelia syndrome 1 patients.

Although OMIM is an important source to investigate the gene associations with diseases, we would like to share our opinions about OMIM and our study. OMIM mainly collected the causal genes for many monogenic and rare diseases. Kidney stone disease is a complex disease that is influenced by many variants or genes with polygenic effects. It's very common that kidney stone disease-associated genes in our GWAS and previous GWASs were not associated with kidney stones (MIM# 167030) in OMIM. For example, the *DGKD* was identified in previous kidney stones GWAS and validated in experiment (Nature Communications 10, 5175 (2019).), however, *DGKD* (MIM * 601826) was not associated with kidney stones in OMIM. Actually, only *SLC26A1* was associated with kidney stones (MIM# 167030) in OMIM.

In addition, the recent PheWAS studies (Nature Reviews Genetics 17, 129-145 (2016)., Nature Genetics 53, 972-981 (2021).) have suggested that some genes

participate in different biological processes and associated with many diseases which can be related or not. Although many candidate genes in our study are associated with Mendelian diseases, the causal variants at these genes may be different between kidney stone disease and these Mendelian diseases. It may be not proper to conclude that the Mendelian diseases which are associated with our GWAS identified genes should have a very clear link to kidney stones or higher risk for kidney stones.

- PM20D1 (line 271-273): Please rephrase this sentence as it sounds like the study that found an association of PM20D1 levels with obesity also found an association with kidney stone disease (KSD), which is not the case. I recommend to make two separate sentences out of this statement for clarification.

Response: Thanks for your suggestion. The sentence has been rephrased as “*PM20D1* is involved in the lipid metabolic process and regulating energy homeostasis. Also serum PM20D1 level was significantly elevated in overweight and obese individuals. It was well known that adiposity increases risk of KSD, and *PM20D1* may result in obesity, thus exacerbating the risk of KSD”.

- SLC41A1: to be critically discussed that the only Mendelian association of the gene refers to a Nephronophthisis-like phenotype but no kidney stone disease (KSD) or nephrocalcinosis (MIM# 619468; PMID: 23661805).

Response: Thanks for your comments. This is a very good clue. Nephronophthisis-like phenotype makes up the most common genetic cause of CKD in early life. The CKD patients have reduced urine pH which promotes the formation of some types of kidney stones (PMID: 37398692). We have added some critical discussion as your suggestion.

- CLDN10: please discuss that patients with the corresponding Mendelian condition (HELIX-Syndrome, MIM# 617671, PMID: 28771254) do not show KSD or nephrocalcinosis.

Response: Thanks for your comments. Actually, the HELIX-Syndrome patients had hypohidrosis, renal loss of NaCl with secondary hyperaldosteronism and hypokalemia, as well as hypolacrymia, ichthyosis, xerostomia, and severe enamel wear (PMID: 28771254). CLDN10 is associated with HELIX-Syndrome for its mutations cause a dysfunction in tight junctions in several tissues, and subsequent abnormalities in renal ion transport, ectodermal gland homeostasis, and epidermal integrity. We think that the

function of CLDN10 in HELIX-Syndrome highlights that CLDN10 was important for regulating transepithelial ion exchange and electrical resistance, which is very important for kidney stone formation. We have added some discussion about the function of CLDN10 in our revised manuscript.

- TRPV5: this is a long-standing candidate gene for hypercalciuria and KSD, as ko mice show severe hypercalciuria (PMID: 14679186) - please cite this reference for completeness and acknowledgement.

- SLC12A1: interesting locus as this is the Bartter Type 1 gene - please add the MIM-number (#601678) as in all Mendelian conditions. Loss-of-function of this gene is not only correlated with hypercalciuria but a very severe (mostly syndromic) disorder associated with nephrocalcinosis and kidney failure - gene discovery publication may be relevant to cite: PMID: 8640224

- SLC30A10: interesting candidate as well - please add MIM# 613280 and reference for Mendelian hypermanganesemia PMID: 22341972.

Response: Thanks for your suggestion. These genes are very good points and we have added it in our revised manuscript.

Of note: the manuscript still needs final editing by a native speaker to enhance both readability and comprehensibility.

Response: Thanks for your suggestion. We have carefully revised the manuscript, and then a native English speaker helped us to polish and improve our manuscript.

Reviewer #2 (Remarks to the Author):

The revisions have improved the manuscript, however there remains a superficiality to the links between identified loci of interest and their function, and it is of some concern that the genes noted to be of interest in the discussion are often only supported by one of two pieces of evidence and potential pathophysiology may not have been fully explored.

Response: We thank the reviewer for acknowledging our effort to improve the manuscript. We have carefully revised the manuscript based on your comments and highlighted the changes in our new manuscript. The remaining comments are addressed below.

For example, PLEKHM1 is highlighted due to its links with osteopetrosis and an analogy drawn with altered BMD in stone formers. However, the study cited concerns osteoporosis in stone formers, which could be considered the opposite phenotype; I think there has been confusion regarding osteopetrosis and osteoporosis.

Response: We apologize for the misuse of osteopetrosis and osteoporosis. *PLEKHM1* has a Mendelian association with osteopetrosis (MIM # 618107). We interpreted the shared pathophysiologic mechanism between osteopetrosis and kidney stones from the role of parathyroid hormones in calcium metabolism. Elevated parathyroid hormones (PTH) have been observed in osteopetrosis patients (PMID 17997709, 27291868). PTH regulates calcium metabolism by increasing the production of 1,25(OH)₂D and calcium reabsorption (PMID 18329892). Notably, the increased 1,25(OH)₂D also has been observed in osteopetrosis patients, which would further increase calcium reabsorption (PMID 27291868). These findings suggest the potential shared regulatory mechanism for osteopetrosis and kidney stones. We have revised the discussion about the potential mechanism in our revised manuscript.

The authors discuss how CYP24A1 may have relevance with regard to vitamin D concentrations, the GWAS summary statistics they have looked to for correlations are based on 25-OH data, CYP24A1 inactivates 1,25-OH vit D, a few steps away from 25-OH, a mechanism which is not noted or discussed.

Response: Thanks for your suggestion. We found that the risk allele T of rs17216707 for kidney stones at *CYP24A1* was significantly associated with higher calcium, and vitamin D concentration. As the metabolic enzyme of 25OHD and 1,25(OH)₂D, loss of

function or reduced activity of *CYP24A1* could result in elevated 25OHD and 1,25(OH)₂D, further resulting in hypercalcemia (New England Journal of Medicine 365, 410-421 (2011)., The Journal of Clinical Endocrinology & Metabolism 100, 2832-2836 (2015).). Our findings suggested the risk allele T of rs17216707 could exert its effect on kidney stones through the dysfunction of *CYP24A1*, which could result in the elevated vitamin D, including both 25OHD and 1,25(OH)₂D, further increasing the reabsorption of calcium. We have added some discussion about the potential mechanism in our revised manuscript.

PM20D1 is highlighted as it has been found to be have increased concentrations in overweight individuals in one study. It is well known that adiposity increases risk of kidney stones. However, it seems a leap to suggest that PM20D1 is responsible for this link, and PM20D1 does not appear to be renally expressed.

Response: Thanks for your comments. We presumed that PM20D1 may result in overweight for its role in the lipid metabolic process and regulating energy homeostasis (Cell 166, 424-435 (2016).), thus increasing the risk kidney stones. Although, *PM20D1* has very low expression in almost all bulk tissues based on the GTEx project, it seems specially highly expressed in a small part of proximal tubule cells based on kidney scRNA-seq profiles (Figure R1). These finding suggested that *PM20D1* may regulate energy homeostasis and then influence the transmembrane transport in proximal tubule. We have added some discussion about the potential mechanism in our revised manuscript.

Figure R1 The expression of *PM20D1* in different cells based on kidney scRNA-seq. The results were downloaded from http://www.susztaklab.com/hk_genemap/scRNA. The scRNA-seq profiles included kidney samples from healthy controls and CKD cases (details in bioRxiv 2022.10.24.513598).

SLC12A1 has in rare cases been linked to Bartters, but isn't the more likely explanation that there is some effect on water reabsorption to affect renal tubular salt concentrations?

Response: Thanks for your suggestion. SLC12A1 encodes a kidney-specific sodium-potassium-chloride cotransporter and functions in concentrating urine and NaCl reabsorption. The corresponding Mendelian condition of SLC12A1 is Bartter syndrome type 1, which is characterized as impaired salt reabsorption with salt wasting, hypokalemic metabolic alkalosis, and hypercalciuria. (OMIM #601678). Specifically, the hypercalciuria was correlated with kidney stones. We have added some discussion about the potential mechanism in our revised manuscript.

I apologise if I missed part of the file download, but I was unable to find the figure legends which has hampered my ability to comment fully on these. I may have misunderstood but in figure 3B your text states:

'Among 220 cell type specific chromatin modification marks, six cell types, including three kidney cell types, were relevant to kidney stone disease' Isn't it the case that there

are six cell specific chromatin modification marks that have been identified rather than six cell types? And within this 3 kidney specific markers?

Response: The comments of the review is right. To make it clearer, we revised the sentence as “Among 220 cell type-specific chromatin modification marks, the specific marks in six cell types, including three kidney cell types, were relevant to kidney stone disease”.

Within the pathway enrichment analysis, the most significantly enriched gene set was “appendage morphogenesis”. What is the explanation for this? Does this maybe suggest that gene prioritisation hasn’t resulted in reliable results as I can't see a link between limb development and stones. In addition, I think GO:0072502 is now obsolete?

Response: Thanks for your comments. We noticed that gene set information in clusterProfiler package we used in our previous analysis did not updated timely. In our revision, we use g:GOST tool implemented in the web server g:Profiler (<https://biit.cs.ut.ee/gprofiler/gost>), which is updated approximately in every three months and follows quarterly releases of Ensembl databases (Nucleic Acids Research 47, W191-W198 (2019).). The most significantly enriched gene sets were “phosphate ion homeostasis” (GO:0055062), “limb morphogenesis” (GO:0035108) and “appendage morphogenesis” (GO:0035107). Previous studies have shown disturbances in phosphate homeostasis can lead to kidney stones (PMID 18293139). Specifically, among the prioritized genes in “limb morphogenesis” and “appendage morphogenesis” pathways, TFAP2B gene is involved in “renal system development” (GO:0072001) and “kidney development” (GO:0001822). ECE1 is involved in “calcitonin catabolic process” (GO:0010816). In addition, MED1 is involved in vitamin D signaling pathway, such as "regulation of vitamin D receptor signaling pathway" (GO:0070562), "cellular response to vitamin D" (GO:0071305), and "response to vitamin D" (GO:0033280). These findings suggested ion homeostasis and kidney-related development play important roles in the formation of kidney stones. We have updated the pathway analysis results and added some discussion in our revised manuscript.

With regard to drug-gene interaction pairs, how have the 38 drugs for renal calculi prevention been defined? I am not clear that there are 38 different drugs available to prevent stones? Is there some overlap between these drugs?

Response: Thanks for your question. We apologize for your confusion. Specifically, we first annotated those genes of interest with respect to known drug-gene interactions and potential druggability by searching in the Drug-Gene Interaction Database (DGIdb, www.dgidb.org). Drugs provided by this database are defined by ChEMBL, which is a manually curated database of bioactive molecules with drug-like properties. Next, we searched the WebMD website (<https://www.webmd.com/drugs>) for medicines that contain the above drug ingredients and can treat kidney stones, which resulted in 38 different drugs (no overlap) that can be composed of medications for kidney stones treatment. We have added more details in the method in our revised manuscript.

In addition, we found a positive but insignificant genetic correlation ($rg=0.33$, $P=0.13$) between kidney stones and vitamin deficiency'. This seems an odd way to report to me, isn't the finding here that there is no association between vit D deficiency and stones?

Response: The comments of the review is right. We did not find significant global genetic correlation between kidney stones and vitamin deficiency, although they share several susceptibility genes in some potential pleiotropic loci. To make it clearer, we revised the sentence as “We found no significant genetic correlation between KSD and vitamin D deficiency, although they share several susceptibility genes in some potential pleiotropic loci”.

The article states that stones can block the ureter to cause infection. It would be my understanding that a urinary infection co-exists with a stone, the blockage means the infection can not adequately clear, it is not the case that the stone causes the infection.

Response: Thank you for the reminder. We agree that the relationship between urinary stones and urinary infections co-exist. The causality is not clear currently. To make it clearer, we revised the sentence as “KSD is also a type of genitourinary disease and always co-exists with kidney infection”.

The snATAC seq presented I think is from mouse tissue rather than human, this should be highlighted. Human scATACseq data is publicly available and would be preferable to use.

Response: Thanks for your question. We carefully checked the data source (<https://www.ncbi.nlm.nih.gov/geo/query/acc.cgi?acc=GSE172008>) and verified that the scATAC-seq data used in our study is from two human kidney samples by 10x

snATAC-seq. To make it clearer, we revised the sentence in method as “We further explored the distribution of kidney stones associated variants in different kidney cell type-specific accessible regions identified by snATAC-seq in two human kidney samples (gene expression omnibus accession number GSE172008).

REVIEWERS' COMMENTS

Reviewer #1 (Remarks to the Author):

The manuscript is further improved. There are only a couple of minor points remaining to address:

-Page 9, line 212: When talking about SLC41A1, the authors write: "The chronic kidney disease patients had reduced urinary pH" this is confusing to me; please clarify this connection.

- Page 11, line 272: "KSD is also a type of genitourinary disease and always co-exists with kidney infection" - I believe the authors confuse inflammation with infection. KSD can surely happen without concomitant infection - please get rid of the word "always" and replace it with "often".

- When reporting Mendelian gene association, the mode of inheritance is missing (monoallelic/heterozygous/dominant versus biallelic/homozygous/compound heterozygous/recessive). Please add this to the text. Example: page 10, line 240: CLDN10 is associated with HELIX syndrome, as its mutations cause ..." - rather "... as biallelic mutations cause ...", another example: page 10, line 254: "Mutations in SLC12A1 were associated with" - rather "Biallelic mutations in SLC12A1"

- Monoallelic alteration of CLDN10 was recently associated with CKD progression (PMID: 36890159) - this may be worth citing as well.

Reviewer #2 (Remarks to the Author):

Thank you for your revisions relating to the use of language:

'Also serum PM20D1 level was significantly elevated in overweight and obese individuals^{34 208}. It was well known that adiposity increases risk of KSD, and PM20D1 may result in obesity, thus exacerbating the risk of KSD.'

Suggest

Serum PM20D1 levels are significantly elevated in overweight and obese individuals. Adiposity is well known to increase risk of kidney stones. Associations of PM20D1 with kidney stones and obesity may indicate a shared causal pathway.

'Additionally, KSD individuals tended to have lower magnesium in their blood than normal people'

Suggest: Additionally, KSD individuals tended to have lower magnesium in their blood than people without kidney stones'

With regard to:

However no genome-wide significant signals ($P < 5 \times 10^{-8}$ ²⁴⁷) associated with KSD were identified at this locus, probably due to the small sample size^{13-16 248}. In this study, we identified a novel signal ²⁴⁹ located 46 bp downstream of the TRPV5 showing an association with KSD (rs4252512, MAF=0.015, OR=0.78, $P = 1.88 \times 10^{-8}$ ²⁵⁰), which may explicate the role of TRPV5 in the ²⁵¹ pathology of KSD.

I am confused as to what the first sentence relates to, do you mean no associations have previously

been identified at this locus?

'KSD is also a type of genitourinary disease and always co-exists with kidney infection5'

This should be revised, suggest

KSD is also a type of genitourinary disease and may co-exist with kidney infection5

Reviewer #1 (Remarks to the Author):

The manuscript is further improved. There are only a couple of minor points remaining to address:

Response: Thank you for your positive comments and previous constructive suggestions for our manuscript. We have addressed the remaining concerns in the revised manuscript. Please find below our point-by-point replies to your comments.

-Page 9, line 212: When talking about SLC41A1, the authors write: "The chronic kidney disease patients had reduced urinary pH" this is confusing to me; please clarify this connection.

Response: Thanks for your comment. The sentence has been revised as "Chronic kidney disease could cause metabolic acidosis, resulting in a reduced urine pH, which could further promote the formation uric acid kidney stones."

- Page 11, line 272: "KSD is also a type of genitourinary disease and always co-exists with kidney infection" - I believe the authors confuse inflammation with infection. KSD can surely happen without concomitant infection - please get rid of the word "always" and replace it with "often".

Response: Thanks for your kind reminder. We have replaced "always" with "may often" in the revised manuscript which also suggested by the 2nd reviewer.

- When reporting Mendelian gene association, the mode of inheritance is missing (monoallelic/heterozygous/dominant versus biallelic/homozygous/compound heterozygous/recessive). Please add this to the text. Example: page 10, line 240: CLDN10 is associated with HELIX syndrome, as its mutations cause ..." - rather "... as biallelic mutations cause", another example: page 10, line 254: "Mutations in SLC12A1 were associated with" - rather "Biallelic mutations in SLC12A1"

Response: Thanks for your suggestion. We have added the mode of inheritance in the revised manuscript, supplementary Table 5, and supplementary Table 8.

- Monoallelic alteration of CLDN10 was recently associated with CKD progression (PMID: 36890159) - this may be worth citing as well.

Response: Thanks for your kind reminder. We have added and discussed the reference in the revised manuscript (page 10, lines 243-245).

Reviewer #2 (Remarks to the Author):

Thank you for your revisions relating to the use of language:

Response: Thank you for your positive comments and previous constructive suggestions for our manuscript. We have addressed the remaining concerns in the revised manuscript. Please find below our point-by-point replies to your comments.

'Also serum PM20D1 level was significantly elevated in overweight and obese individuals. It was well known that adiposity increases risk of KSD, and PM20D1 may result in obesity, thus exacerbating the risk of KSD.'

Suggest

Serum PM20D1 levels are significantly elevated in overweight and obese individuals. Adiposity is well known to increase risk of kidney stones. Associations of PM20D1 with kidney stones and obesity may indicate a shared causal pathway.

Response: Thank you for your kind suggestion. We have replaced the sentence in the revised manuscript.

'Additionally, KSD individuals tended to have lower magnesium in their blood than normal people'

Suggest:

Additionally, KSD individuals tended to have lower magnesium in their blood than people without kidney stones'

Response: Thank you for your kind suggestion. We have replaced the sentence in the

revised manuscript.

With regard to: However no genome-wide significant signals ($P < 5 \times 10^{-8}$) associated with KSD were identified at this locus, probably due to the small sample size¹³⁻¹⁶. In this study, we identified a novel signal located 46 bp downstream of the TRPV5 showing an association with KSD (rs4252512, MAF=0.015, OR=0.78, $P=1.88 \times 10^{-8}$), which may explicate the role of TRPV5 in the pathology of KSD. I am confused as to what the first sentence relates to, do you mean no associations have previously been identified at this locus? '

Response: The reviewer is right. To make it clear, we revised the sentences as “However, no association with $P < 5 \times 10^{-8}$ has previously been identified at this locus, probably due to the small sample size”.

KSD is also a type of genitourinary disease and always co-exists with kidney infection⁵

This should be revised,

suggest

KSD is also a type of genitourinary disease and may co-exist with kidney infection

Response: Thank you for your kind suggestion. We have replaced the sentence in the revised manuscript.